# STAMBPL1 activates the GRHL3/HIF1A/VEGFA axis through interaction with FOXO1 to promote angiogenesis in triple-negative breast cancer

Huan Fang[1,2†], Huichun Liang[1†], Chuanyu Yang[1], Dewei Jiang[1], Qianmei Luo[1], Wen-Ming Cao[3]*, Huifeng Zhang[4]*, Ceshi Chen[1,5,6]*

[1]Kunming Institute of Zoology, Chinese Academy of Sciences, Kunming, China; [2]Kunming College of Life Sciences, University of Chinese Academy of Sciences, Kunming, China; [3]The Department of Breast Medical Oncology, Zhejiang Cancer Hospital, Hangzhou, China; [4]Department of Pharmacy, the First People's Hospital of Yunnan Province/the Affiliated Hospital of Kunming University of Science and Technology, Kunming, China; [5]Academy of Biomedical Engineering, Kunming Medical University, Kunming, China; [6]The Third Affiliated Hospital, Kunming Medical University, Kunming, China

*For correspondence:
caowm@zjcc.org.cn (W-MC);
zhanghuifeng92@163.com (HZ);
chenc@kmmu.edu.cn (CC)

†These authors contributed equally to this work

Competing interest: The authors declare that no competing interests exist.

## eLife Assessment

The study conducted by Fang et al. offers significant and **fundamental** insights, notably enhancing our understanding of angiogenesis. While some of the claims are supported by **convincing** experimental approaches, others lack sufficient validation. Additionally, there are instances where critical experimental controls appear to be absent.

**Abstract** In the clinic, anti-tumor angiogenesis is commonly employed for treating recurrent, metastatic, drug-resistant triple-negative, and advanced breast cancer. Our previous research revealed that the deubiquitinase STAMBPL1 enhances the stability of MKP-1, thereby promoting cisplatin resistance in breast cancer. In this study, we discovered that STAMBPL1 could upregulate the expression of the hypoxia-inducible factor HIF1α in breast cancer cells. Therefore, we investigated whether STAMBPL1 promotes tumor angiogenesis. We demonstrated that STAMBPL1 increased *HIF1A* transcription in a non-enzymatic manner, thereby activating the HIF1α/VEGFA signaling pathway to facilitate triple-negative breast cancer angiogenesis. Through RNA-seq analysis, we identified the transcription factor GRHL3 as a downstream target of STAMBPL1 that is responsible for mediating *HIF1A* transcription. Furthermore, we discovered that STAMBPL1 regulates *GRHL3* transcription by interacting with the transcription factor FOXO1. These findings shed light on the role and mechanism of STAMBPL1 in the pathogenesis of breast cancer, offering novel targets and avenues for the treatment of triple-negative and advanced breast cancer.

## Introduction

Triple-negative breast cancer (TNBC) is a subtype of breast cancer characterized by a lack of expression of estrogen receptor, progesterone receptor, and human epidermal growth factor receptor 2. Despite representing only 10–15% of all breast cancer cases, it is associated with a greater risk of recurrence,

metastasis, and resistance to chemotherapy, leading to a poorer prognosis than other types of breast cancer (*Geyer et al., 2017*). Therefore, understanding the pathogenesis of this subtype of breast cancer and identifying effective treatment targets are key areas of research focus and challenge. In the clinic, in addition to surgery, radiotherapy, and chemotherapy, treatment approaches for TNBC also include the use of epidermal growth factor receptor (EGFR) inhibitors, poly (ADP-ribose) polymerase (PARP) inhibitors, immune checkpoint inhibitors, and anti-angiogenic therapies (*Zagami and Carey, 2022*; *Garcia et al., 2020*).

For solid tumors, the survival, proliferation, and invasion of cancer cells rely on surrounding blood vessels to provide nutrients and oxygen. Cancer cells can increase the synthesis and secretion of the vascular endothelial growth factor A (VEGFA), activate endothelial cells in neighboring blood vessels through paracrine pathways, and stimulate tumor angiogenesis (*Detmar, 2000*). Therefore, investigating the upstream molecular mechanisms that activate VEGFA in cancer cells offers new targets for inhibiting TNBC angiogenesis via disruption of this signaling pathway.

The deubiquitinase STAMBPL1 (also known as AMSH-LP) belongs to the AMSH family and cleaves K63-linked polyubiquitin chains (*Sato et al., 2008*). STAMBPL1 stabilizes XIAP to inhibit apoptosis in prostate cancer cells (*Chen et al., 2019*) and confers resistance to honokiol-induced apoptosis in various cancer cell types by stabilizing Survivin and c-FLIP (*Woo et al., 2019*). Our previous research demonstrated that STAMBPL1 enhances cisplatin resistance in TNBC cells through the stabilization of MKP-1 (*Liu et al., 2022*). In this study, we discovered that, independent of its deubiquitinating enzyme activity, STAMBPL1 upregulates HIF1α expression in TNBC cells. We aimed to investigate the molecular mechanism by which it upregulates HIF1α and explore its role in tumor angiogenesis in TNBC.

FOXO1, a member of the forkhead box transcription factor family, regulates various physiological and pathological processes, such as cell proliferation, apoptosis, autophagy, and oxidative stress (*Xing et al., 2018*). In breast cancer, FOXO1 enhances the stemness of cancer cells by promoting *SOX2* transcription (*Yu et al., 2019*) and confers resistance to chemotherapy drugs in basal-like breast cancer cells by activating *KLF5* transcription (*Cui et al., 2024*). Furthermore, FOXO1 plays a crucial role in angiogenesis by upregulating *VEGFA* transcription (*Jeon et al., 2018*). However, the specific mechanism by which FOXO1 regulates *VEGFA* transcription remains unclear, and its role in tumor angiogenesis in TNBC requires further investigation.

In this study, we discovered that STAMBPL1 facilitates the transcriptional regulation of *GRHL3* by interacting with FOXO1, consequently enhancing *HIF1A* transcription through GRHL3 to activate the HIF1α/VEGFA pathway. This results in increased endothelial cell activity via paracrine signaling, thereby promoting tumor angiogenesis in TNBC.

## Methods
### Cell lines and reagents
All cell lines used in this study, including HCC1806, HCC1937, and HEK293T cells, were purchased from ATCC (American Type Culture Collection, Manassas, VA, USA) and validated by short tandem repeat analysis, and these cell lines tested negative for mycoplasma contamination. HCC1806 and HCC1937 cells were cultured in RPMI 1640 medium supplemented with 5% FBS. HEK293T cells were cultured in DMEM (Thermo Fisher, Grand Island, USA) with 5% FBS at 37°C with 5% $CO_2$. Primary human umbilical vein endothelial cells (HUVECs) were maintained in an EGM-2 Bullet Kit (CC-3162, Lonza, USA). AS1842856 (Cat#HY-100596) and apatinib (Cat#HY-13342S) were purchased from MCE (New Jersey, USA).

### Plasmid construction and stable STAMBPL1, GRHL3, and FOXO1 overexpression
We constructed the full-length *STAMBPL1/GRHL3/FOXO1* genes and then subcloned them into the pCDH lentiviral vector. The packaging plasmids (pMDLg/pRRE, pRSV-Rev, and pCMV-VSV-G) and the pCDH-STAMBPL1/GRHL3/FOXO1 expression plasmid were cotransfected into HEK293T cells ($2 \times 10^6$ in 10 cm plates) to produce lentiviruses. Following transfection for 48 hr, the lentivirus was collected and used to infect HCC1806 and HCC1937 cells. Forty-eight hours later, puromycin (2 µg/ml) was used to screen the cell populations.

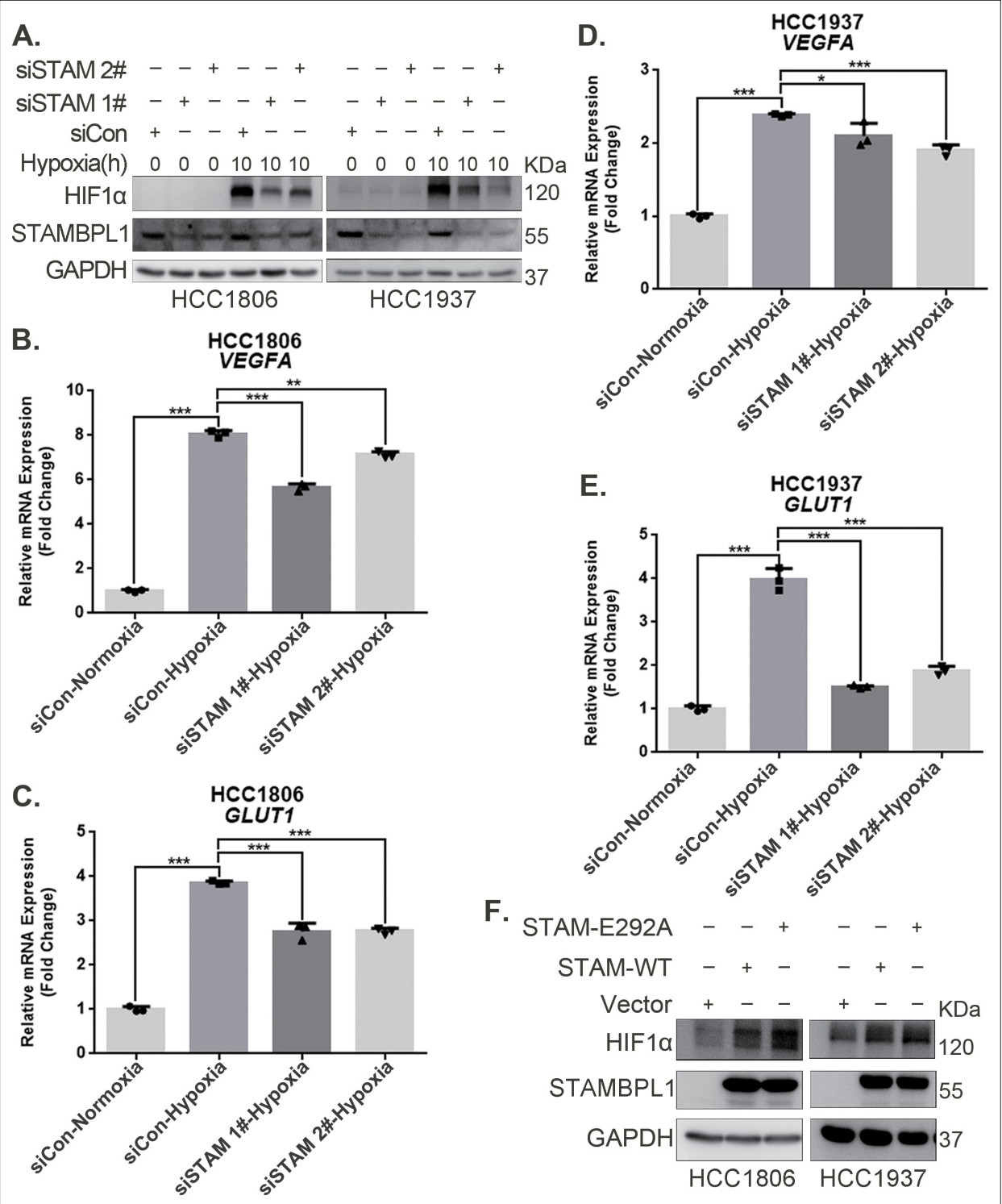

**Figure 1.** STAMBPL1 upregulates HIF1α in a non-enzymatic manner in triple-negative breast cancer (TNBC) cells. (**A**) Western blot detected the protein levels of HIF1α when STAMBPL1 was knocked down under normoxia and hypoxia for 10 hr. (**B, C**) RNA samples were collected after 10 h of hypoxia treatment following the knockdown of STAMBPL1 in HCC1806 cells. RT-qPCR experiments were performed to detect the mRNA levels of the HIF1α downstream targets *VEGFA* and *GLUT1* (n=3). (**D, E**) RNA samples of HCC1937 were collected as the above HCC1806 cells. RT-qPCR experiments were performed to detect the mRNA levels of the HIF1α downstream targets *VEGFA* and *GLUT1* (n=3). (**F**) Western blot detected the protein levels of HIF1α when STAMBPL1/STAMBPL1-E292A was overexpressed under normoxia in HCC1806 and HCC1937 cells. Mean ± SD, *p < 0.05, **p < 0.01, ***p < 0.001, *t*-test.

*Figure 1 continued on next page*

*Figure 1 continued*

The online version of this article includes the following source data for figure 1:

**Source data 1.** PDF file containing original western blots for *Figure 1A, F*, indicating the relevant bands and treatments.

**Source data 2.** Original files for western blot analysis displayed in *Figure 1A, F*.

## Stable knockdown of STAMBPL1 and GRHL3

The pSIH1-H1-puro shRNA vector was used to express STAMBPL1, GRHL3 and luciferase (LUC) shRNAs. *STAMBPL1* shRNA#1, 5′-GGAGCATCAGAGATTGATA-3′; *STAMBPL1* shRNA#2, 5′-GCTG CTATGCCTGACCATA-3′; *GRHL3* shRNA#1, 5′-CCTTGAGCTTCCTCTATGA-3′; *GRHL3* shRNA#2, 5′-AGAGGAAGATGCGCGATGA-3′; *Luciferase* shRNA, 5′-CUUACGCUGAGUACUUCGA-3′; HCC1806 and HCC1937 cells were infected with lentivirus. Stable populations were selected via the use of 1–2 mg/ml puromycin. The knockdown effect was evaluated by western blotting.

## RNA interference

The siRNA target sequences used in this study were as follows: *STAMBPL1* siRNA#1, 5′-GGAGCATC AGAGATTGATA-3′; *STAMBPL1* siRNA#2, 5′-GCTGCTATGCCTGACCATA-3′; *GRHL3* siRNA#1, 5′-CCTT GAGCTTCCTCTATGA-3′; *GRHL3* siRNA#2, 5′-AGAGGAAGATGCGCGATGA-3′; *HIF1α* siRNA#1, 5′-AAGAGGTGGATATGTCTGG-3′; *HIF1α* siRNA#2, 5′-CGTCGAAAAGAAAAGTCTCTT-3′; *FOXO1* siRNA#1, 5′-CCCAGAUGCCUAUACAAAC-3′; and *FOXO1* siRNA#2, 5′-CTCAAATGCTAGTACTATTA G-3′. All siRNAs were synthesized by RiboBio (RiboBio, China) and transfected at a final concentration of 50 nM.

## WB and antibodies

The WB procedure was described in our previous study (*Chen et al., 2005*). Anti-STAMBPL1 (sc-376526) and anti-GAPDH (sc-25778) antibodies were purchased from Santa Cruz Biotechnology (Santa Cruz, CA, USA). Anti-FOXO1 (#2880S) and anti-HIF1α (#36169S) antibodies were purchased from CST. Anti-β-actin (A5441) and anti-GST (G7781) antibodies were purchased from Sigma-Aldrich (St. Louis, MO, USA). The anti-Flag (ab205606) antibody was purchased from Abcam.

## Real-time polymerase chain reaction

Total RNA was extracted via TRIzol (Invitrogen, 15596026). One microgram of total RNA was reverse transcribed to cDNA according to the manufacturer's instructions for the HiScript II QRT SuperMix for qPCR Kit (Vazyme, R223-01). For quantitative PCR (RT-qPCR), the SYBR Green Select Master Mix system (Applied Biosystems, 4472908, USA) was used on an ABI-7900HT system (Applied Biosystems, 4351405). The primers used for PCR were as follows: 18S forward, 5′-CTCAACACGGGAAACCTCAC -3′; 18S reverse, 5′-CGCTCCACCAACTAAGAACG-3′; HIF1α forward, 5′-AAGTCTGCAACATGGA AGGTAT-3′; HIF1α reverse, 5′-TGAGGAATGGGTTCACAAATC-3′; VEGFA forward, 5′-TGAGGAAT GGGTTCACAAATC-3′; VEGFA reverse, 5′-ATCTGCATGGTGATGTTGGA-3′; GLUT1 forward, 5′-TCGT CGGCATCCTCATCGCC-3′; GLUT1 reverse, 5′-CCGGTTCTCCTCGTTGCGGT-3′; GRHL3 forward, 5′-GGGGCTGAGGAATGCGATCT-3′; and GRHL3 reverse, 5′-AATTTTGCCGTCCAGCTCCC-3′.

## Cell proliferation and migration assays

To detect the proliferation of HUVECs, we used the Click-iT EdU Alexa Fluor 647 Imaging Kit (Invitrogen) according to the manufacturer's protocol. Briefly, HUVECs were seeded on coverslips (BD Biosciences) at $0.5 \times 10^5$ cells per well. The next day, the supernatants were discarded, and the cells were cultured with conditioned medium (CM). Six hours later, the cells were incubated with EdU in CM for 4 hr, followed by fixation and staining. For each sample, three random fields were observed via fluorescence microscopy, and the total numbers of cells and EdU-positive cells were counted. To detect the migration of HUVECs, we performed a wound-healing assay. Twenty-four hours after seeding, the supernatants of the HUVECs were discarded, and the cells were scratched and cultured with CM for 24 hr. Wound closure was imaged via microscopy. For each image, the gap width was analyzed via ImageJ.

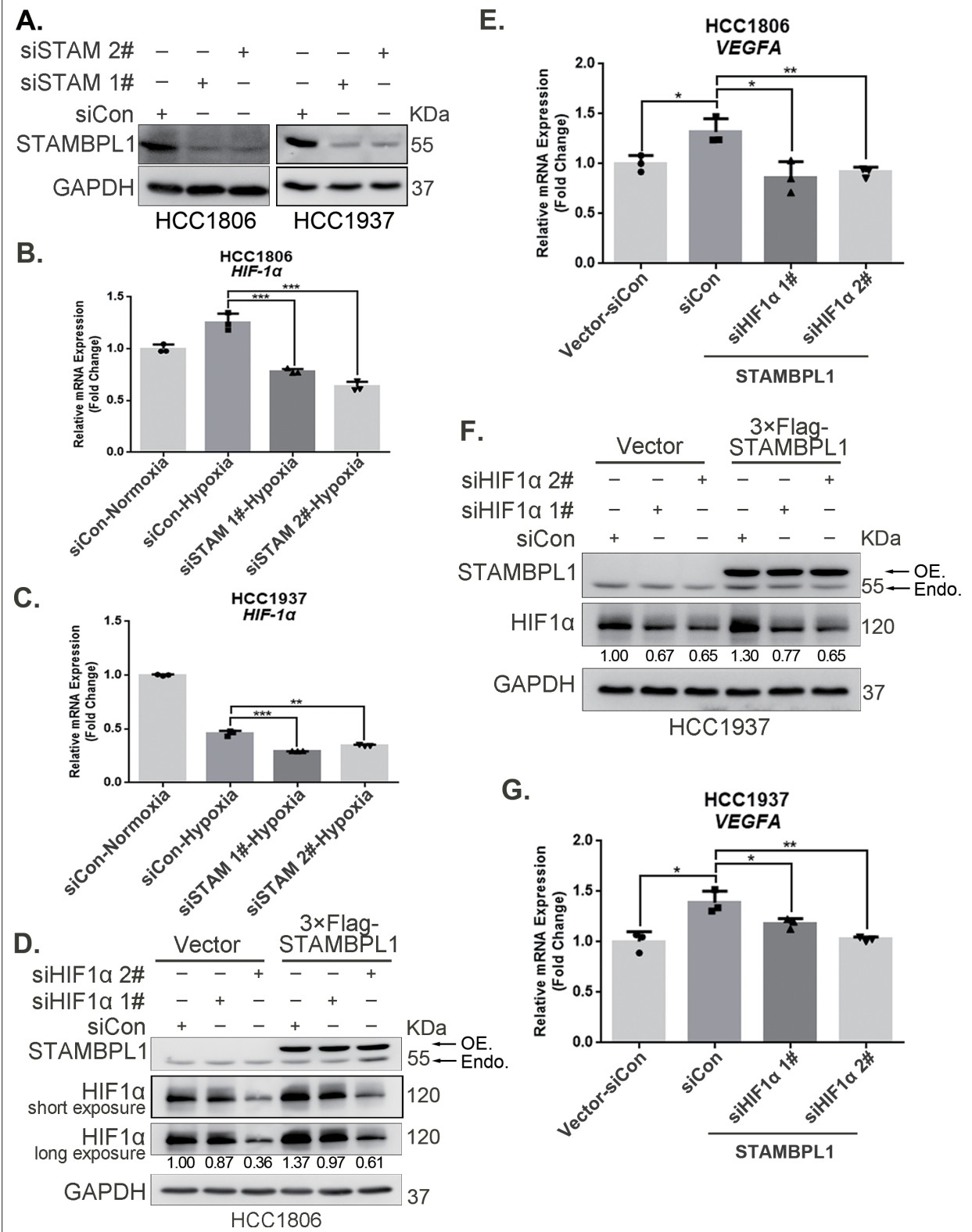

**Figure 2.** STAMBPL1 promotes *HIF1A* transcription and activates the HIF1α/VEGFA axis. (**A**) Western blot analysis was performed to confirm the knockdown of STAMBPL1 in HCC1806 and HCC1937 cells. (**B**) RNA samples were collected after 10 hr of hypoxia treatment following the knockdown of STAMBPL1 in HCC1806 cells. RT-qPCR experiments were performed to detect the mRNA levels of HIF1α (n = 3). (**C**) RNA samples of HCC1937 cells were collected as the above HCC1806 cells. RT-qPCR experiments were performed to detect the mRNA levels of HIF1α (n = 3). (**D**) Knocking down HIF1α

*Figure 2 continued on next page*

*Figure 2 continued*

using siRNAs in HCC1806 cells stably overexpressing STAMBPL1 under normoxia. Western blot was used to detect the protein levels of STAMBPL1 and HIF1α. (**E**) RT-qPCR experiments were performed to detect the effect of HIF1α knockdown on the mRNA level of *VEGFA* mRNA in HCC1806 cells with STAMBPL1 overexpression under normoxia (n = 3). (**F**) Knocking down HIF1α using siRNAs in HCC1937 cells stably overexpressing STAMBPL1 under normoxia. Western blot was used to detect the protein levels of STAMBPL1 and HIF1α. (**G**) RT-qPCR experiments were performed to detect the effect of HIF1α knockdown on the mRNA level of *VEGFA* mRNA in HCC1937 cells with STAMBPL1 overexpression under normoxia (n = 3). Mean ± SD, *p < 0.05, **p < 0.01, ***p < 0.001, *t*-test.

The online version of this article includes the following source data for figure 2:

**Source data 1.** PDF file containing original western blots for *Figure 2A, D, F*, indicating the relevant bands and treatments.

**Source data 2.** Original files for western blot analysis displayed in *Figure 2A, D, F*.

## Tube formation assays

HUVECs (1 × 10$^4$) in CM were seeded onto Matrigel (BD Biosciences)-coated μ-Slide angiogenesis plates (ibidiGmbH, Munich, Germany). At 6 hr after seeding, images were taken via microscopy and then analyzed with ImageJ. The total branch length was measured.

## RNA-seq analysis

Total RNA was isolated from HCC1806 cells (5 × 10$^5$ cells per sample) following treatment with either control (NC) or STAMBPL1 siRNA using TRIzol reagent (Thermo Fisher Scientific, Cat#15596026). The extracted RNA was subsequently subjected to commercial RNA sequencing analysis conducted by Wuhan Aiji Baike Biotechnology Co, Ltd (Wuhan, China). Differentially expressed genes were identified based on the following criteria: log$_2$[Fold Change] <−1.0 with a statistical significance threshold of p-value <0.05. The source data of RNA-seq could be found in *Figure 4—source data 3*.

## Chromatin immunoprecipitation assays

After the sample preparation was completed, the subsequent experimental steps were performed according to the instructions of the Simple ChIP (R) Plus Kit (CST, # 9005). The PCR primers for amplifying the region of interest on the HIF1α gene promoter were as follows: 5′-GACTGACAGGCTTGAA GTTTATGC-3′ and 5′-TGTTGCTGTAAACTTCAAGGGAAA-3′, and the PCR primers for amplifying the region of interest on the GRHL3 gene promoter were as follows: 5′-TTCTATCCCTTCTGTGCTGA CCA-3′ and 5′-TGTGCCAGACCCTACTCTGGG-3′.

## Dual-luciferase reporter assays

The DNA fragments HIF1α and GRHL3 were amplified from MCF10A cell genomic DNA via PCR template and then cloned and inserted into the pGL3-Basic vector. HEK293T cells were seeded in 24-well plates and transfected with pCDH-GRHL3-3×Flag or pCDH-FOXO1-3×Flag and pGL3 luciferase reporter plasmids (both 600 ng/well) together with the pCMV-Renilla control (5 ng/well). After transfection for 48 hr, the cell lysates were collected, and the luciferase activities were detected via the dual-luciferase reporter assay system (Promega, USA). For the WT HIF1α promoter (with the GRHL3-binding motif), the primers used for PCR were as follows: forward, 5′-gctagcccgggctcgagatctCCAC TGCGCTCCAGCCTG-3′; reverse, 5′-cagtaccggaatgccaagcttCCTCAGACGAGGCAGCACTG-3′. For the mutant HIF1α promoter (without the GRHL3-binding motif), the primers used for PCR were as follows: forward, 5′-TCTTTCCCTGAGGCCTTCCTATATGCTTAT-3′; reverse, 5′-ATAAGCATATAGGAAG GCCTCAGGGAAAGA-3′. For the WT GRHL3 promoter (with the FOXO1-binding motif), the primers used for PCR were as follows: forward, 5′-gctagcccgggctcgagatctATTAACAAGGGTGACTGAAG AGGG-3′; reverse, 5′-cagtaccggaatgccaagcttTGGAGGTATACCTCAACAGGTGC-3′. For the mutant GRHL3 promoter (without the FOXO1-binding motif), the primers used for PCR were as follows: forward, 5′-CTCCCCCACCAAACAAAGAAGGAGAACACCCC-3′; reverse, 5′-GGGGTGTTCTCCTTCT TTGTTTGGTGGGGGAG-3′.

## Immunofluorescence staining

HEK293T cells plated on cell culture slides were transfected with the STAMBPL1 and FOXO1 expression plasmids. Two days after transfection, the cells were fixed in 3.7% polyformaldehyde at 4°C overnight. After being blocked with 5% BSA at room temperature for 1 hr, the cells were stained

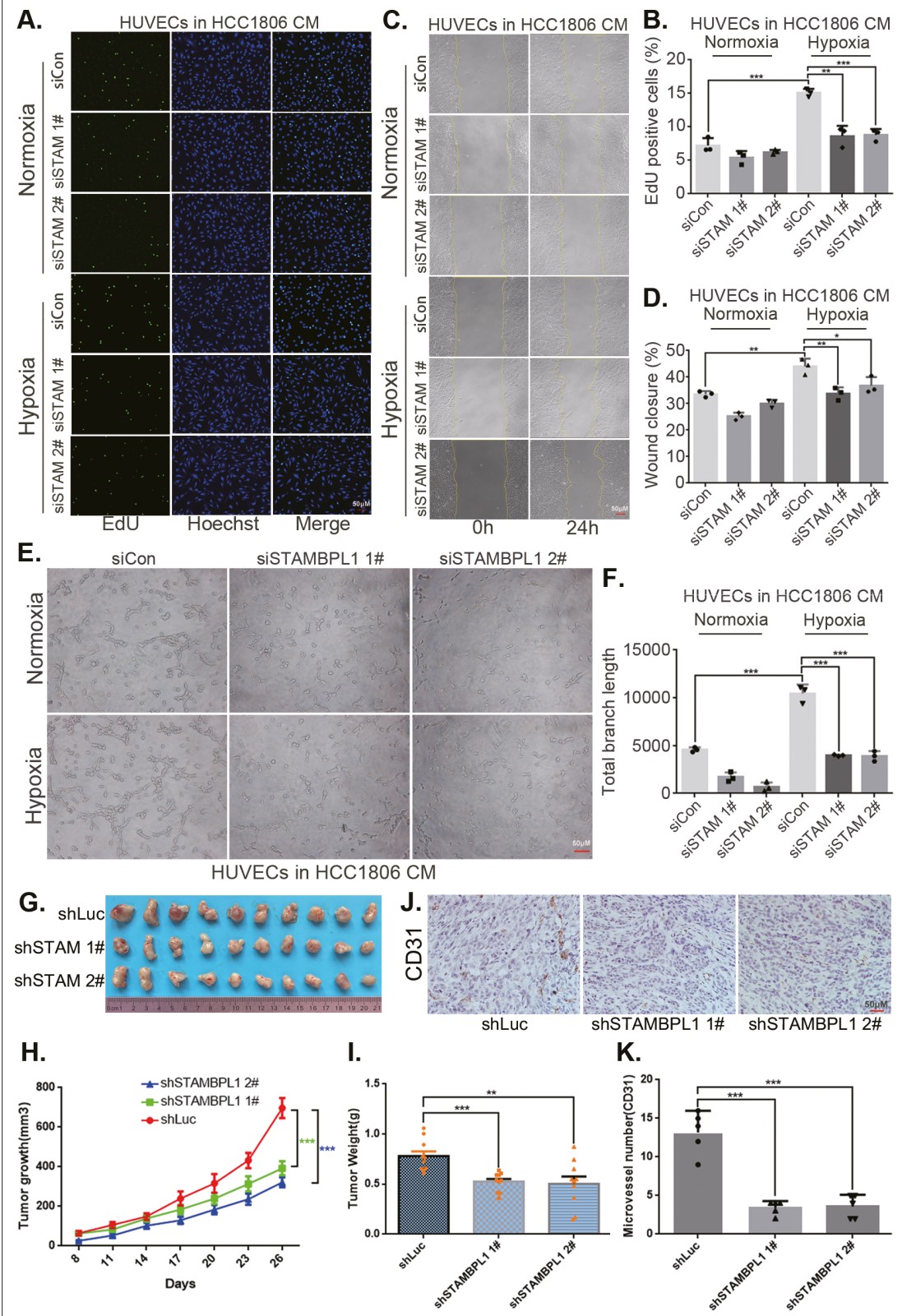

**Figure 3.** STAMBPL1 in triple-negative breast cancer (TNBC) cells enhances the activity of human umbilical vein endothelial cells (HUVECs) and promotes TNBC angiogenesis. (**A**) EDU assay was performed to detect the effect of HCC1806 conditioned medium (CM) with STAMBPL1 knockdown under normoxia or hypoxia on the proliferation of HUVECs. The scale bar means 50 μm. (**B**) Statistical analysis of the EdU assay results (n = 3). The scale bar means 50 μm.(**C**) Wound healing assay was performed to detect the effect of HCC1806 CM with STAMBPL1 knockdown under normoxia

*Figure 3 continued on next page*

*Figure 3 continued*

or hypoxia on the migration of HUVECs. (**D**) Statistical analysis of the wound healing assay results (n = 3). (**E**) Tube formation assay was performed to detect the effect of HCC1806 CM with STAMBPL1 knockdown under normoxia or hypoxia on the tube formation of HUVECs. The scale bar means 50 µm. (**F**) Statistical analysis of the tube formation assay results (n = 3). (**G**) The growth of TNBC xenograft tumors was evaluated by photographing the tumors in nude mice to assess the role of STAMBPL1. (**H, I**) The growth and weight of the transplanted tumors in the nude mice were statistically analyzed, and the STAMBPL1sh#1 and STAMBPL1sh#2 groups were compared with the shLuc group (n = 10). (**J**) An immunohistochemical assay was used to detect the expression of the angiogenesis marker CD31 in xenograft tumors. The scale bar means 50 µm. (**K**) The number of microvessels in the immunohistochemical experiments was statistically analyzed (n = 5). Mean ± SD, *p < 0.05, **p < 0.01, ***p < 0.001, *t*-test.

The online version of this article includes the following figure supplement(s) for figure 3:

**Figure supplement 1.** STAMBPL1 in triple-negative breast cancer (TNBC) cells enhances the activity of human umbilical vein endothelial cells (HUVECs).

**Figure supplement 2.** STAMBPL1 in triple-negative breast cancer (TNBC) cells enhances the activity of human umbilical vein endothelial cells (HUVECs).

with anti-STAMBPL1 (mouse) and anti-FOXO1 (rabbit) antibodies at 4°C overnight. The cells were subsequently stained with both an Alexa Fluor647-labeled anti-mouse secondary antibody and a FITC-labeled anti-rabbit secondary antibody (ZSGB-Bio, Beijing, China) at room temperature for 1 hr. Nuclei were stained with DAPI (Biosharp, BL739A) for 15 min. Images were captured via a confocal microscope.

## Immunoprecipitation and GST pull-down

For endogenous protein interaction, cell lysates were first incubated with anti-FOXO1 antibodies or rabbit IgG (2729S; Cell Signaling Technology) and then incubated with protein A/G magnetic beads (HY-K0202; MCE). For the GST pull-down assay, the cell lysates were directly incubated with Glutathione Sepharose 4B (10312185; Cytiva) overnight at 4°C. For the IP-Flag assay, cell lysates were directly incubated with anti-Flag magnetic beads (HY-K0207; MCE) overnight at 4°C. The precipitates were washed four times with 1 ml of lysis buffer, boiled for 10 min with 1× SDS sample buffer, and subjected to WB analysis.

## Xenograft experiments

We purchased 5- to 6-week-old female BALB/c nude mice from SLACCAS (Changsha, China). Animal feeding and experiments were approved by the animal ethics committee of Kunming Institute of Zoology, Chinese Academy of Sciences. HCC1806-shLuc, HCC1806-shSTAMBPL1, or HCC1806-shGRHL3 cells and HCC1806-PCDH-Vector or HCC1806-PCDH-STAMBPL1 cells (1 × 10$^6$ in Matrigel (BD Biosciences, NY, USA)) were implanted into the mammary fat pads of the mice. When the tumor volume reached approximately 50 mm$^3$, the nude mice were randomly assigned to the control or treatment groups (n = 4/group). The control group was given vehicle alone, and the treatment group received the FOXO1 inhibitor AS1842856 (10 mg/kg), the VEGFR inhibitor apatinib (50 mg/kg), and the FOXO1 inhibitor AS1842856 combined with the VEGFR inhibitor apatinib via intragastric administration every 2 days for 20 days. The tumor volume was calculated as follows: tumor volume was calculated by the formula ($\pi$ × length × width$^2$)/6. The maximal tumor size permitted by the animal ethics committee of Kunming Institute of Zoology, Chinese Academy of Sciences, and the maximal tumor size in this study was not exceeded.

## Immunohistochemical staining

The xenograft tumor tissues were fixed in 3.7% formalin solution. The immunohistochemistry was performed on 4-µm-thick paraffin sections after pressure-cooking for antigen retrieval. An anti-CD31 primary antibody (1:400, Abcam, ab28364) was used. After 12 hr, the slides were washed three times with PBS and incubated with secondary antibodies (hypersensitive enzyme-labeled goat anti-mouse/rabbit IgG polymer (OriGene, China) at room temperature for 20 min, DAB concentrated chromogenic solution (1:200 dilution of concentrated DAB chromogenic solution)), counterstained with 0.5% hematoxylin, dehydrated with graded concentrations of ethanol for 3 min each (70–80–90–100%), and finally stained with dimethyl benzene. The immunostained slides were evaluated via light microscopy, and the number of microvessels with positive CD31 expression was counted.

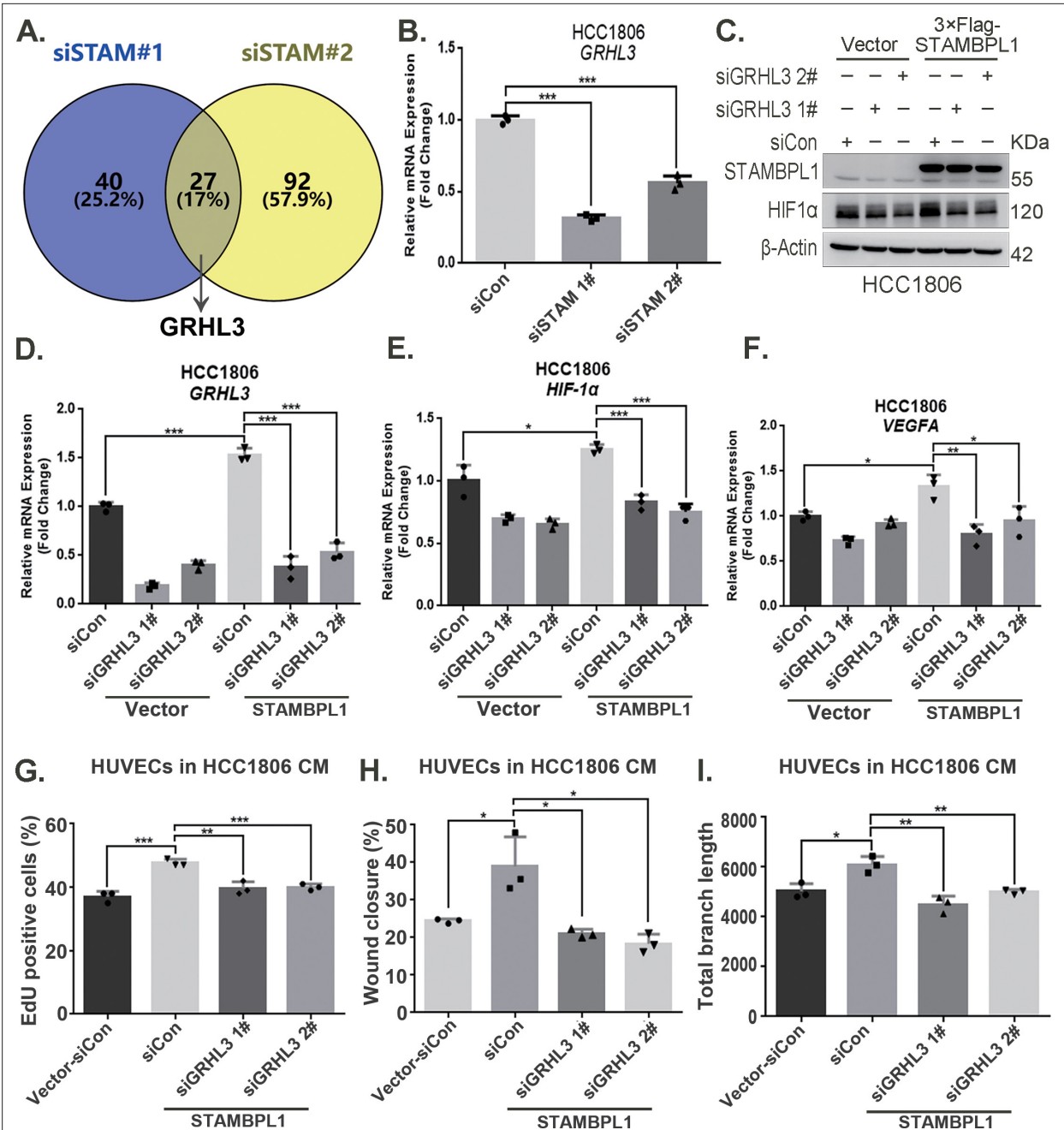

**Figure 4.** STAMBPL1 promotes *HIF1A* transcription via the upregulation of GRHL3. (**A**) Venn diagram showing the overlap of differentially expressed genes identified via RNA-seq analysis. (**B**) RT-qPCR experiment was performed to detect the effect of STAMBPL1 knockdown on the mRNA levels of GRHL3 in HCC1806 cells (n = 3). (**C**) In HCC1806 cells stably overexpressing STAMBPL1, GRHL3 was knocked down via siRNA, after which the protein was collected. Western blotting was performed to detect the effect of GRHL3 knockdown on the expression of HIF1α as STAMBPL1 overexpression under normoxia. (**D–F**) RT-qPCR experiments were performed to detect the knockdown efficiency of GRHL3 and the effect of GRHL3 knockdown on the mRNA levels of HIF1α and its downstream target vascular endothelial growth factor A (VEGFA) when STAMBPL1 was overexpressing (n = 3). (**G**) Statistical analysis of the EdU assay results (n = 3). (**H**) Statistical analysis of the wound healing assay results (n = 3). (**I**) Statistical analysis of the tube formation assay results (n = 3). Mean ± SD, *p < 0.05, **p < 0.01, ***p < 0.001, *t*-test.

The online version of this article includes the following source data and figure supplement(s) for figure 4:

**Source data 1.** PDF file containing original western blots for *Figure 4C*, indicating the relevant bands and treatments.

**Source data 2.** Original files for western blot analysis displayed in *Figure 4C*.

**Source data 3.** Original files for RNA-seq analysis displayed in *Figure 4A*.

*Figure 4 continued on next page*

*Figure 4 continued*

**Figure supplement 1.** STAMBPL1 promotes *HIF1A* transcription via upregulating GRHL3.

**Figure supplement 1—source data 1.** PDF file containing original western blots for *Figure 4—figure supplement 1C*, indicating the relevant bands and treatments.

**Figure supplement 1—source data 2.** Original files for western blot analysis displayed in *Figure 4—figure supplement 1C*.

## Statistical analysis

All the graphs were created via GraphPad Prism software version 8.0. Comparisons between two independent groups were assessed via two-tailed Student's *t*-tests. One-way analysis of variance with least significant differences was used for multiple group comparisons. p values of <0.05, 0.01, or 0.001 were considered to indicate statistically significant results, and comparisons that were significant at the 0.05 level are indicated by *, those at the 0.01 level are indicated by **, and those at the 0.001 level are indicated by ***.

## Results

### STAMBPL1 upregulates HIF1α in a non-enzymatic manner in TNBC cells

Our previous research revealed that STAMBPL1, a deubiquitinase, promotes cisplatin resistance in TNBC by stabilizing MKP1 (*Liu et al., 2022*). To further investigate the role and mechanism of STAMBPL1 in TNBC, we discovered that the knockdown of STAMBPL1 can inhibit hypoxia-induced HIF1α expression (*Figure 1A*) and suppress the transcription of the HIF1α downstream genes *VEGFA* and *GLUT1* (*Figure 1B–E*). Under normoxic conditions, both STAMBPL1 and its enzymatically mutated variant (E292A) can upregulate the protein expression of HIF1α (*Figure 1F*). These findings suggest that STAMBPL1 enhances the expression of HIF1α in breast cancer cells through a non-DUB enzyme activity mechanism.

### STAMBPL1 promotes *HIF1A* transcription and activates the HIF1α/VEGFA axis

To elucidate how STAMBPL1 upregulates HIF1α, we examined the transcription levels of *HIF1A* when STAMBPL1 was knocking down under hypoxia. These results indicate that knocking down STAMBPL1 significantly inhibits the transcription of *HIF1A* (*Figure 2A–C*). At the same time, we detected the transcription level of *HIF1A* and *VEGFA* when STAMBPL1 was overexpressing under normoxia. Data showed that STAMBPL1 overexpression increased the transcription of *HIF1A* (Figure 4E, Figure 4—figure supplement 1E) and *VEGFA* (*Figure 2E, G*). The ability of STAMBPL1 to induce *VEGFA* transcription was blocked by HIF1α knockdown (*Figure 2D–G*). These findings suggested that STAMBPL1 activated the HIF1α/VEGFA axis through enhancing the transcription of *HIF1A*.

### STAMBPL1 in TNBC cells enhances the activity of HUVECs and promotes TNBC angiogenesis

The CM from TNBC cells with STAMBPL1 knockdown inhibited the proliferation (*Figure 3A, B*, *Figure 3—figure supplement 1A, B*), migration (*Figure 3C, D*, *Figure 3—figure supplement 1C, D*), and tube formation (*Figure 3E, F*, *Figure 3—figure supplement 1E, F*) of HUVECs. When STAMBPL1 was overexpressed in TNBC cells, the CM of HCC1806 and HCC1937 cells promoted the ability of HUVECs to proliferate (*Figure 3—figure supplement 2A, D*), migrate (*Figure 3—figure supplement 2B, E*), and form tubes (*Figure 3—figure supplement 2C, F*), which could be reversed by knocking down HIF1α in TNBC cells. These findings suggest that STAMBPL1 activates the HIF1α/VEGFA axis in TNBC cells, leading to enhanced abilities of HUVECs through a paracrine pathway. Knocking down STAMBPL1 inhibited the growth of HCC1806 xenografts in mice (*Figure 3G–I*) and decreased the number of microvessels in tumor tissue (*Figure 3J, K*), indicating that STAMBPL1 promoted tumor angiogenesis.

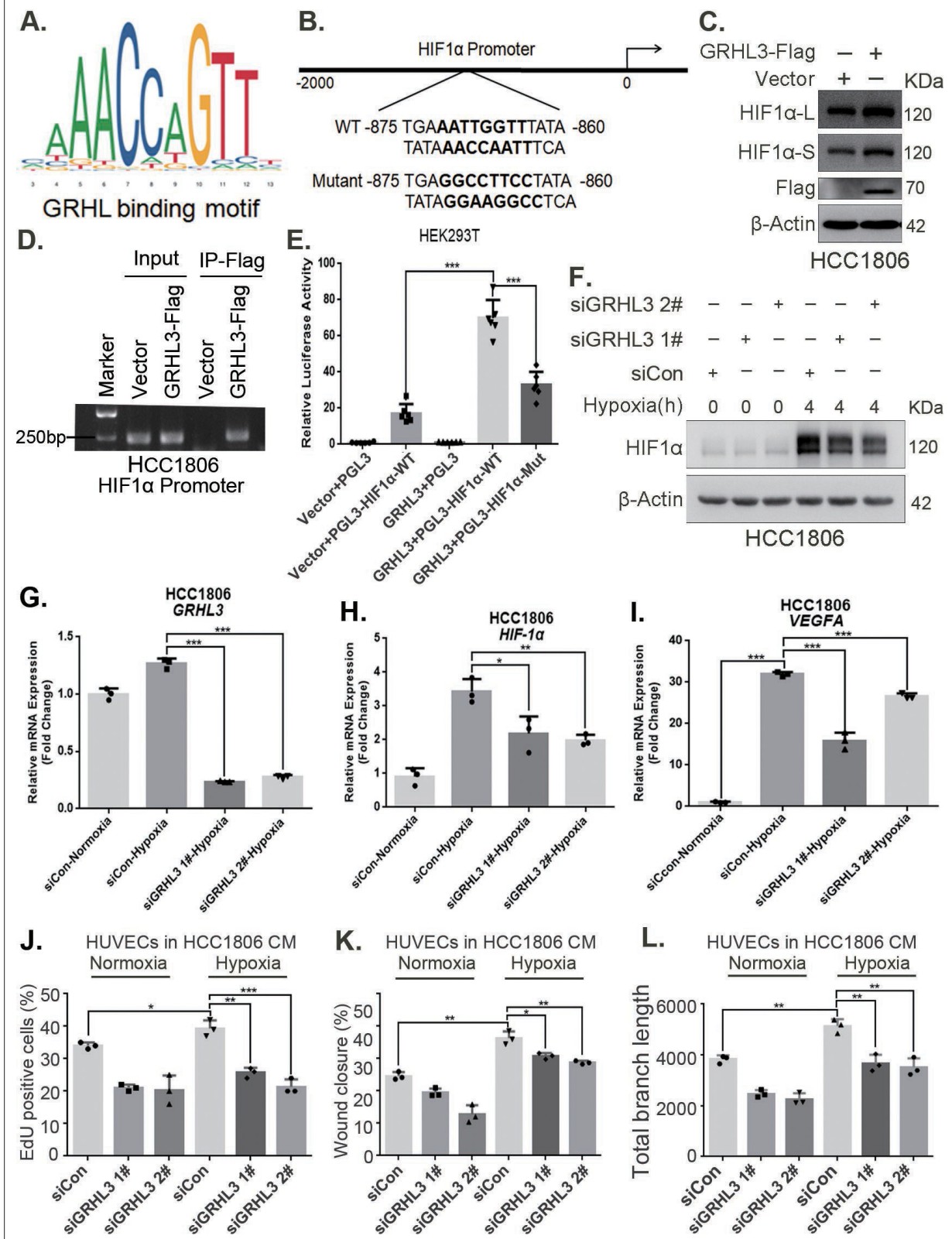

**Figure 5.** GRHL3 enhances *HIF1A* transcription by binding to its promoter. (**A**) The JASPAR website was used to predict the potential binding sequence of the transcription factor GRHL3 to the HIF1α promoter. (**B**) Mutation pattern of the HIF1α promoter-binding sequence. (**C**) The protein level of HIF1α was detected in HCC1806 cells with GRHL3 overexpression by using western blot. (**D**) ChIP–PCR experiments conducted in HCC1806 cells stably overexpressing GRHL3 to detect the interaction between GRHL3 and the promoter of *HIF1A* gene. (**E**) A luciferase assay was performed in HEK293T

*Figure 5 continued on next page*

*Figure 5 continued*

cells to detect the effect of GRHL3 on the transcriptional activity of *HIF1A* promoter (n = 6). (**F**) In HCC1806 cells, GRHL3 was knocked down by siRNA, and the cells were subjected to hypoxia for 4 hr. Western blotting was performed to detect the effect of GRHL3 knockdown on the protein level of HIF1α. (**G–I**) RT-qPCR experiments were used to detect the effect of GRHL3 knockdown on the mRNA levels of HIF1α and its downstream target vascular endothelial growth factor A (VEGFA) (n = 3). (**J**) In HCC1806 cells, GRHL3 was knocked down via siRNA, and the cells were then subjected to hypoxia for 24 hr. The conditioned medium (CM) was collected and used to treat human umbilical vein endothelial cells (HUVECs). EdU assays were performed to detect the effect of GRHL3 knockdown CM on the proliferation of HUVECs. Statistical analysis of the EdU assay results was performed (n = 3). (**K**) Wound healing assays were performed to detect the effect of GRHL3 knockdown CM on the migration of HUVECs. Statistical analysis of the wound healing assay results was performed (n = 3). (**L**) The tube formation assay was performed to detect the effect of GRHL3 knockdown CM on the tube formation of HUVECs. Statistical analysis of the tube formation assay results was performed (n = 3). Mean ± SD, *p < 0.05, **p < 0.01, ***p < 0.001, *t*-test.

The online version of this article includes the following source data and figure supplement(s) for figure 5:

**Source data 1.** PDF file containing original western blots and DNA gels for *Figure 5C, D, F*, indicating the relevant bands and treatments.

**Source data 2.** Original files for western blot and DNA gel analysis displayed in *Figure 5C, D, F*.

**Figure supplement 1.** GRHL3 enhances *HIF1A* transcription by binding to its promoter.

**Figure supplement 1—source data 1.** PDF file containing original western blots for *Figure 5—figure supplement 1A*, indicating the relevant bands and treatments.

**Figure supplement 1—source data 2.** Original files for western blot analysis displayed in *Figure 5—figure supplement 1A*.

## STAMBPL1 promotes *HIF1A* transcription via the upregulation of GRHL3

By silencing the *STAMBPL1* gene in HCC1806 cells subjected to 10 hr of hypoxia, we performed RNA-seq analysis to investigate the mechanism by which STAMBPL1 promotes *HIF1A* transcription. We found that silencing of STAMBPL1 resulted in the downregulation of 27 genes (of which only 18 were annotated). Of these 18 genes, except for GRHL3, which is a transcription factor reported to be involved in gene transcription regulation, the remaining 17 genes have no documented association with RNA transcription, stability, or modification (*Figure 4A*, *Figure 4—figure supplement 1A*). Silencing of STAMBPL1 inhibited *GRHL3* transcription in both HCC1806 and HCC1937 cells (*Figure 4B*, *Figure 4—figure supplement 1B*). Conversely, the overexpression of STAMBPL1 promoted *GRHL3* transcription (*Figure 4D*, *Figure 4—figure supplement 1D*). Furthermore, the stimulatory effects of STAMBPL1 on the HIF1α/VEGFA axis (*Figure 4C–F*, *Figure 4—figure supplement 1C–F*) and on HUVECs (*Figure 4G–I*, *Figure 4—figure supplement 1G–I*) were reversed by GRHL3 knockdown. These findings suggest that STAMBPL1 activates the HIF1α/VEGFA axis by upregulating GRHL3.

## GRHL3 enhances *HIF1A* transcription by binding to its promoter

On the basis of the GRHL-binding motif (*Figure 5A*), a GRHL-binding sequence was identified in the *HIF1A* promoter (*Figure 5B*). ChIP and luciferase assays revealed that GRHL3 bound to the *HIF1A* promoter (*Figure 5D*) and increased its activity (*Figure 5E*). Knockdown of GRHL3 (*Figure 5G*, *Figure 5—figure supplement 1B*) resulted in decreased expression of HIF1α at both the mRNA (*Figure 5H*, *Figure 5—figure supplement 1C*) and protein levels (*Figure 5F*, *Figure 5—figure supplement 1A*), leading to suppressed *VEGFA* transcription (*Figure 5I*, *Figure 5—figure supplement 1D*), indicating that GRHL3 promotes *HIF1A* transcription by binding to its promoter.

Furthermore, CM from TNBC cells with GRHL3 knockdown inhibited HUVEC proliferation, migration, and tube formation (*Figure 5J–L*, *Figure 5—figure supplement 1E–G*). The overexpression of GRHL3 in TNBC cells activated the HIF1α/VEGFA axis (*Figure 6A, B*, *Figure 6—figure supplement 1A–C*), resulting in increased proliferation, migration, and tube formation in HUVECs. This effect was reversed by knocking down HIF1α in TNBC cells (*Figure 6C–E*, *Figure 6—figure supplement 1D,E*). Knockdown of GRHL3 (*Figure 6F*) also inhibited tumor growth in HCC1806 xenograft mice (*Figure 6G–I*) and reduced the number of microvessels in tumor tissues (*Figure 6J, K*).

## STAMBPL1 mediates *GRHL3* transcription by interacting with FOXO1

Our previous study demonstrated that STAMBPL1 is localized in the nucleus (*Liu et al., 2022*). This protein may promote the transcription of GRHL3 through interactions with other transcription factors. Previous studies have indicated that FOXO1 acts as an upstream transcription factor of GRHL3 (*Nagarajan et al., 2017*). Therefore, we aimed to investigate whether STAMBPL1 promotes GRHL3

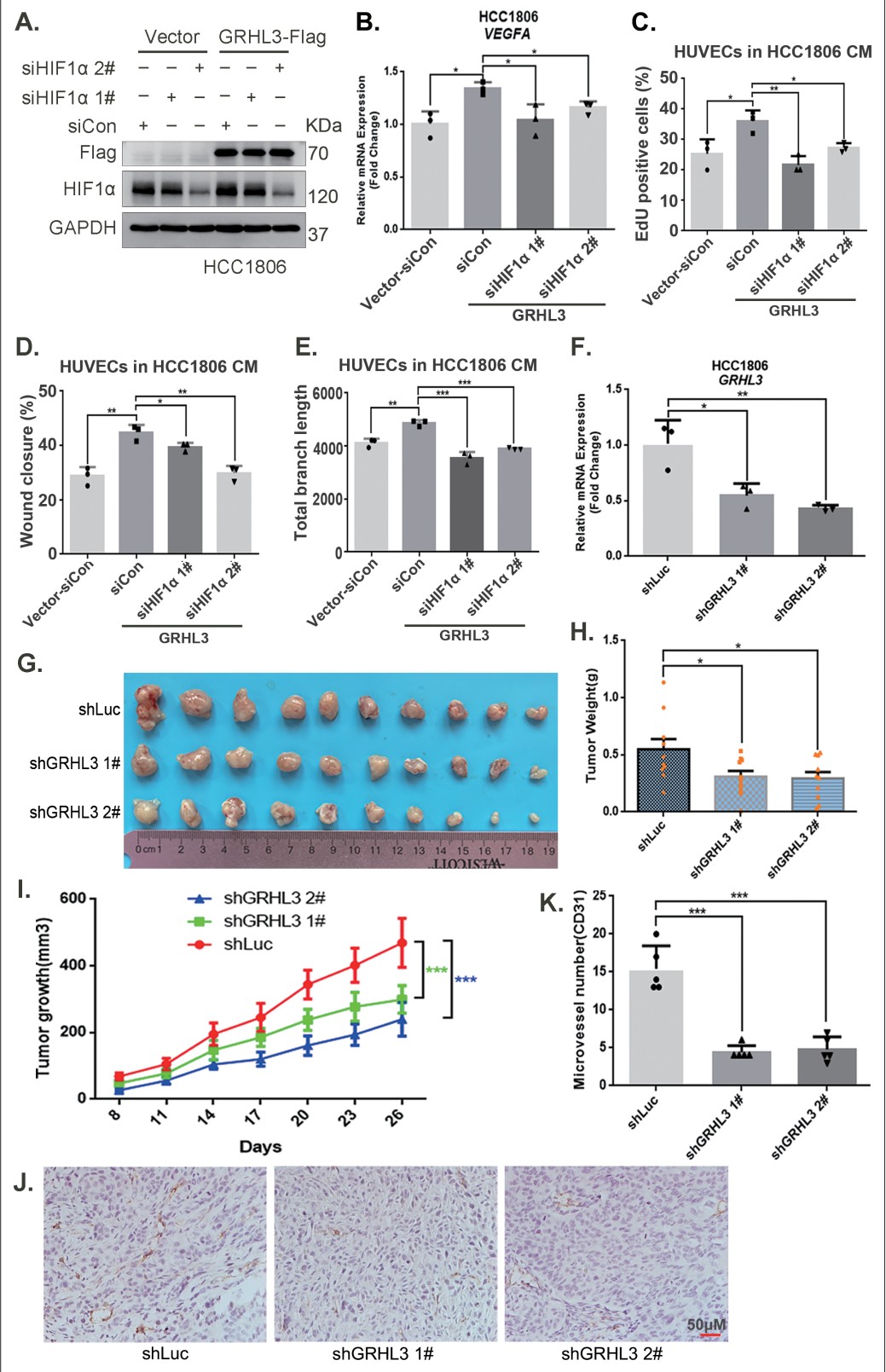

**Figure 6.** GRHL3 enhances *HIF1A* transcription by binding to its promoter. (**A**) In HCC1806 cells stably overexpressing GRHL3, HIF1α was knocked down by siRNA, and the protein was subsequently collected. Western blotting was performed to detect the protein level of GRHL3-Flag and HIF1α. (**B**) RT-qPCR experiments were performed to detect the effect of HIF1α knockdown on the mRNA level of vascular endothelial growth

*Figure 6 continued on next page*

*Figure 6 continued*

factor A (VEGFA) as GRHL3 overexpression (n = 3). (**C**) HCC1806 cells stably overexpressing GRHL3 were used for the knockdown of HIF1α via siRNA. The conditioned medium (CM) was then collected and used to treat human umbilical vein endothelial cells (HUVECs). EdU assay was performed to detect the effect of CM with HIF1α knockdown on the proliferation of HUVECs as GRHL3 overexpression (n = 3). (**D**) Wound healing assay was performed to detect the effect of CM with HIF1α knockdown on the migration of HUVECs as GRHL3 overexpression (n = 3). (**E**) The tube formation assay was performed to detect the effect of CM with HIF1α knockdown on the tube formation of HUVECs as GRHL3 overexpression (n = 3). (**F**) RT-qPCR was used to detect the knockdown of GRHL3 in HCC1806 cells (n = 3). (**G**) Photographs of xenograft tumors from nude mice were taken to evaluate the role of GRHL3 in the growth of triple-negative breast cancer (TNBC) xenograft tumors. (**H**) The weights of the transplanted tumors from the nude mice were statistically analyzed, and the shGRHL3 1# and shGRHL3 2# groups were compared with the shLuc group (n = 10). (**I**) The growth of transplanted tumors in nude mice was statistically analyzed, and the shGRHL3 1# and shGRHL3 2# groups were compared with the shLuc group (n = 10). (**J**) An immunohistochemical assay was performed to detect the expression of the angiogenesis marker CD31 in xenograft tumors. The scale bar means 50 μm. (**K**) Statistical analysis of the number of microvessels in the immunohistochemical experiments (n = 5). Mean ± SD, *p < 0.05, **p < 0.01, ***p < 0.001, *t*-test.

The online version of this article includes the following source data and figure supplement(s) for figure 6:

**Source data 1.** PDF file containing original western blots for *Figure 6A*, indicating the relevant bands and treatments.

**Source data 2.** Original files for western blot analysis displayed in *Figure 6A*.

**Figure supplement 1.** GRHL3 enhances *HIF1A* transcription by binding to its promoter.

**Figure supplement 1—source data 1.** PDF file containing original western blots for *Figure 6—figure supplement 1A*, indicating the relevant bands and treatments.

**Figure supplement 1—source data 2.** Original files for western blot analysis displayed in *Figure 6—figure supplement 1A*.

transcription via FOXO1. Our experimental data revealed that knockdown of FOXO1 in TNBC cells not only inhibited the GRHL3/HIF1α/VEGFA axis (*Figure 7A–D*, *Figure 7—figure supplement 1A–D*) but also reversed the stimulatory effects of STAMBPL1 on this axis (*Figure 7E–H*). Furthermore, FOXO1 was found to bind to the promoter region of GRHL3 (*Figure 7I*, *Figure 7—figure supplement 1F*) and enhance its transcriptional activity (*Figure 7J*). STAMBPL1 was shown to increase the transcriptional activation of FOXO1 at the GRHL3 promoter (*Figure 7K*), and it also bound to the GRHL3 promoter which could be disrupted by FOXO1 knockdown (*Figure 7L*). However, the overexpression of STAMBPL1 and its enzymatic activity mutants did not affect the protein expression level of FOXO1 (*Figure 7—figure supplement 1E*). To investigate the mechanism by which STAMBPL1 promotes GRHL3 transcription through FOXO1, we conducted immunofluorescence and immunoprecipitation assays. We observed the colocalization of STAMBPL1 and FOXO1 in the nucleus (*Figure 7M*) and identified an interaction between them in HCC1937 cells (*Figure 7N*). Through the generation of a series of Flag-FOXO1/GST-fused STAMBPL1 deletion mutants, we mapped the regions of the proteins responsible for this interaction (*Figure 7—figure supplement 1G-H*). The results of the GST pull-down assay indicated that the N-terminus (1–140 aa) of STAMBPL1 interacted with FOXO1 (*Figure 7—figure supplement 1I*). Additionally, an immunoprecipitation assay revealed an interaction between the FOXO1 protein (250–596 aa) and STAMBPL1 (*Figure 7—figure supplement 1J*). These findings suggest that STAMBPL1 promotes GRHL3 transcription by interacting with FOXO1.

## The combination of VEGFR and FOXO1 inhibitors synergistically suppresses TNBC xenograft growth

Through an analysis of the Metabric database in BCIP (http://www.omicsnet.org/bcancer), it was observed that while the expression levels of STAMBPL1, FOXO1, and GRHL3 in breast cancer tissues are not universally elevated compared to adjacent non-cancerous tissues, but their expression levels in TNBC are significantly higher than those in non-TNBC (*Figure 8A–C*, *Figure 8—figure supplement 1A–C*). To assess the potential therapeutic efficacy of cotreatment with the FOXO1 inhibitor AS1842856 and the VEGFR inhibitor apatinib in vivo, we conducted animal experiments in nude mice. HCC1806 cells overexpressing STAMBPL1 were orthotopically implanted into the mammary fat pads of 6-week-old female mice (*n* = 16/group). Once the tumor volume reached approximately 50 mm³, the

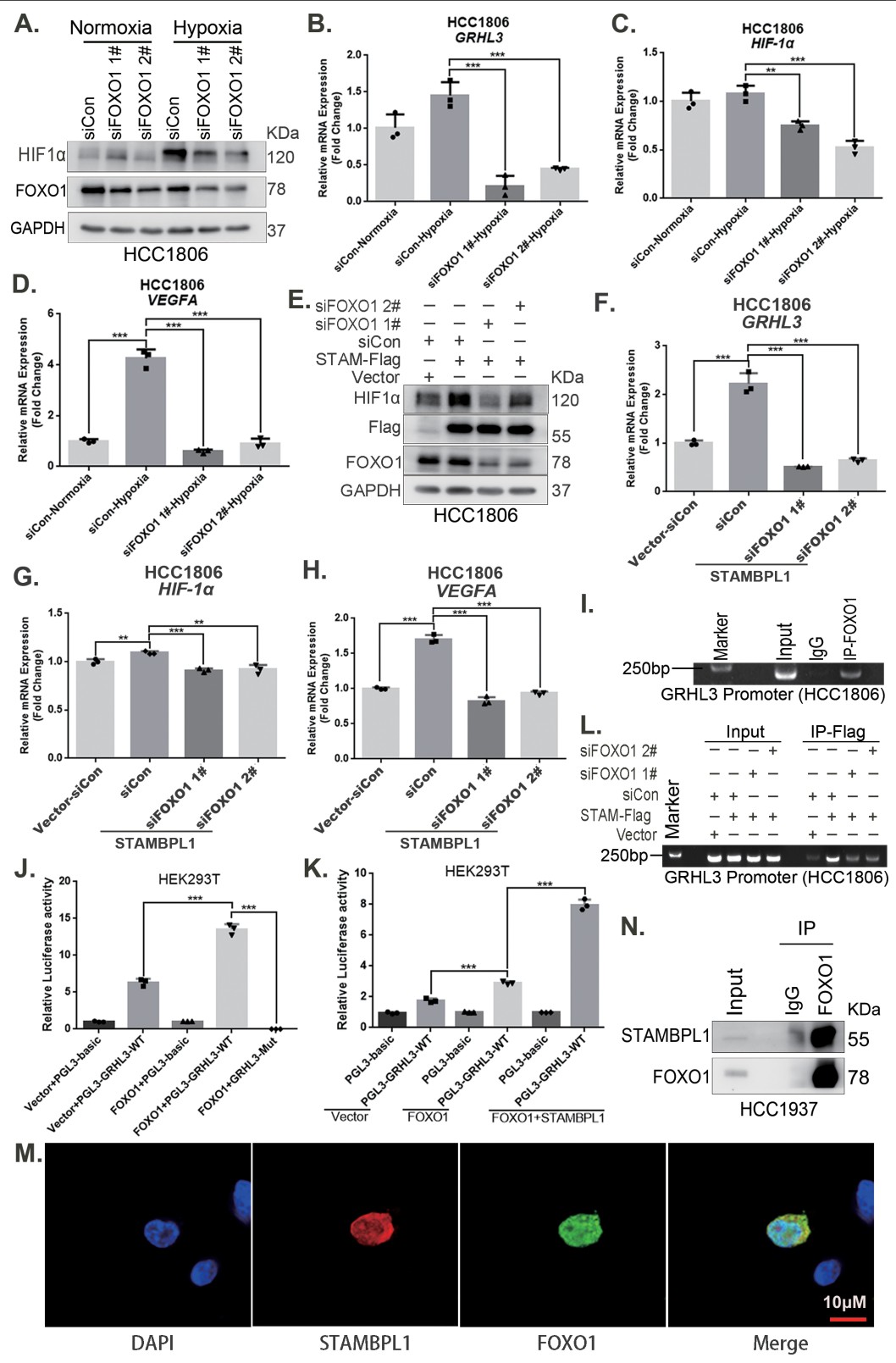

**Figure 7.** STAMBPL1 mediates *GRHL3* transcription by interacting with FOXO1. (**A**) Western blotting was used to detect the HIF1α protein expression when FOXO1 was knocking down in HCC1806 cells followed by hypoxia treatment for 4 hr. (**B–D**) RT-qPCR experiments were performed to detect the effect of FOXO1 knockdown on the mRNA levels of GRHL3/HIF1α/VEGFA in HCC1806 cells under hypoxia for 4 hr. (**E**) In HCC1806 cells stably

*Figure 7 continued on next page*

*Figure 7 continued*

overexpressing STAMBPL1, protein samples were collected after FOXO1 was knocked down via siRNA. Western blotting was performed to detect the effect of FOXO1 knockdown on the expression of HIF1α as STAMBPL1 overexpression. (F–H) RT-qPCR experiments were performed to detect the effect of FOXO1 knockdown on the mRNA expression of GRHL3, HIF1α and VEGFA as STAMBPL1 overexpression. (I) An endogenous ChIP–PCR assay was performed using an anti-FOXO1 antibody in HCC1806 cells. (J) A luciferase assay was performed in HEK293T cells to detect the effect of FOXO1 on the transcriptional activity of GRHL3 promoter. (K) A luciferase assay was conducted in HEK293T cells to detected the effect of STAMBPL1 on the activation of the GRHL3 promoter by FOXO1. (L) ChIP–PCR experiments were performed in HCC1806 cells stably overexpressing STAMBPL1 after knocking down FOXO1 via siRNA. (M) PCDH-STAMBPL1−3×Flag and PCDH-FOXO1−3×Flag plasmids were co-transfected into HEK293T cells, and then, immunofluorescence experiments were performed. Red represents STAMBPL1 staining, green represents FOXO1 staining, and blue represents DAPI staining. (N) Endogenous STAMBPL1 protein was immunoprecipitated from HCC1937 cells via an anti-FOXO1 antibody. Immunoglobulin G (IgG) served as the negative control. Endogenous STAMBPL1 was detected via western blotting.The scale bar means 10 µm. n = 3, Mean ± SD, **p < 0.01, ***p < 0.001, *t*-test.

The online version of this article includes the following source data and figure supplement(s) for figure 7:

**Source data 1.** PDF file containing original western blots and DNA gels for *Figure 7A, E, I, L, N*, indicating the relevant bands and treatments.

**Source data 2.** Original files for western blot and DNA gel analysis displayed in *Figure 7A, E, I, L, N*.

**Figure supplement 1.** STAMBPL1 mediates *GRHL3* transcription by interacting with FOXO1.

**Figure supplement 1—source data 1.** PDF file containing original western blots for *Figure 7—figure supplement 1A, E, I, J*, indicating the relevant bands and treatments.

**Figure supplement 1—source data 2.** Original files for western blot analysis displayed in *Figure 7—figure supplement 1A, E, I, J*.

---

mice were divided into four subgroups to receive apatinib (50 mg/kg, once every 2 days), AS1842856 (10 mg/kg, once every 2 days), a combination of both drugs, or vehicle control for 20 days. Our findings indicated that STAMBPL1 overexpression enhanced breast cancer cell growth in vivo. While the individual inhibitory effects of the FOXO1 and VEGFR inhibitors on tumor growth were not significant, the combined treatment markedly suppressed tumor growth in nude mice (*Figure 8E–G*). Importantly, the drug treatments did not affect the body weights of the mice (*Figure 8H*). These results suggest that the combined administration of AS1842856 and apatinib effectively inhibits tumor growth in nude mice.

## Discussion

In this study, we discovered that STAMBPL1 promotes angiogenesis in TNBC by activating the HIF1α/VEGFA pathway independently of its enzymatic activity. Through RNA-seq analysis, we revealed that STAMBPL1 positively regulates the transcription of *GRHL3*. Furthermore, we demonstrated that GRHL3 binds to the promoter of the *HIF1A* gene, thereby increasing its transcription. Additionally, we found that STAMBPL1 interacts with FOXO1 to facilitate the transcription of GRHL3/HIF1α/VEGFA. Importantly, we confirmed that the combination of the FOXO1 inhibitor AS1842856 and the VEGFR inhibitor apatinib effectively inhibited tumor growth in nude mice. These findings suggest that both STAMBPL1 and FOXO1 may be potential therapeutic targets for inhibiting angiogenesis in TNBC. This study is the first to uncover the role of STAMBPL1 in promoting angiogenesis through the FOXO1/GRHL3/HIF1α axis. Furthermore, a novel transcription factor, GRHL3, which regulates the transcription of HIF1α was identified. These findings provide valuable insights for developing new therapeutic strategies to target TNBC (*Figure 9*).

The role of STAMBPL1 in tumors has not been fully recognized. Recent studies have reported its involvement in the epithelial–mesenchymal transition of various cancers, and its absence has been shown to affect the mesenchymal phenotype of lung and breast cancer (*Ambroise et al., 2020*). Our previous research demonstrated that STAMBPL1 can stabilize MKP1 through deubiquitination and that the deletion of STAMBPL1 and MKP1 increases the sensitivity of breast cancer cells to cisplatin (*Liu et al., 2022*), suggesting that STAMBPL1 may be a potential therapeutic target for breast cancer. However, the mechanism by which STAMBPL1 activates the transcriptional activity of FOXO1 has not

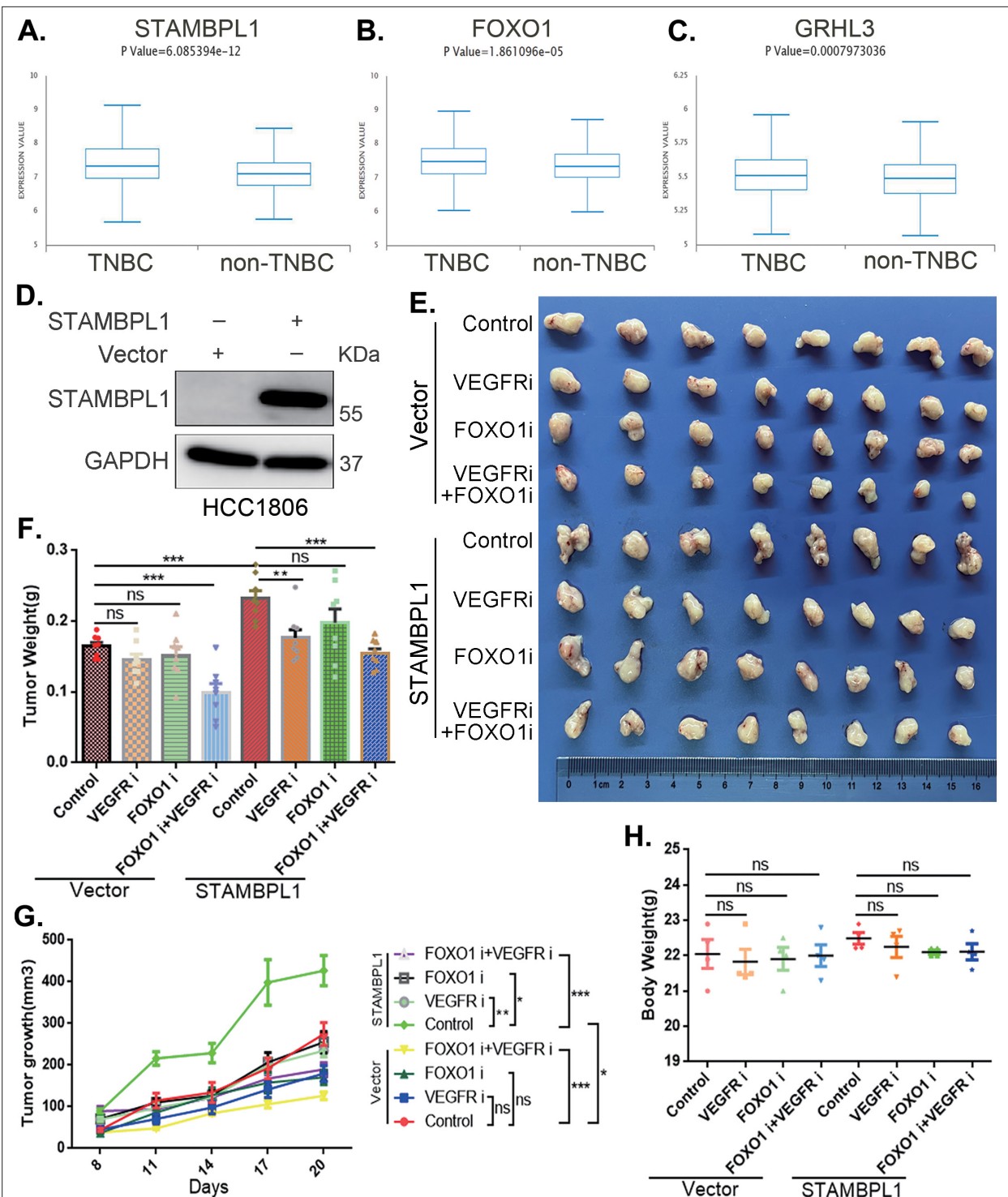

**Figure 8.** The combination of VEGFR and FOXO1 inhibitors synergistically suppresses triple-negative breast cancer (TNBC) xenograft growth. (**A–C**) Metabric database analysis in BCIP revealed high expression levels of STAMBPL1, FOXO1, and GRHL3 in TNBC (*n* = 319) compared with non-TNBC (*n* = 1661). (**D**) The effect of STAMBPL1 overexpression in HCC1806 cells was detected by Western blotting. (**E**) Nude mice with tumors received the FOXO1 inhibitor AS1842856 (10 mg/kg, every 2 days) or the VEGFR inhibitor Apatinib (50 mg/kg, every 2 days) or a combination of both treatments. The effect of the drug treatments on the transplanted tumors was assessed by imaging the tumors. (**F, G**) The weight and growth of the transplanted tumors in the nude mice were statistically analyzed. The vector control group and the STAMBPL overexpression group were compared. The nondrug group and the combined drug group in the vector control group were compared. The nondrug group and the combined drug group in the STAMBPL overexpression

*Figure 8 continued on next page*

*Figure 8 continued*

group were compared (n = 8). (**H**) The final weights of the nude mice were statistically analyzed (n = 4). Mean ± SD, *p < 0.05, **p < 0.01, ***p < 0.001, *t*-test.

The online version of this article includes the following source data and figure supplement(s) for figure 8:

**Source data 1.** PDF file containing original western blots for *Figure 8D*, indicating the relevant bands and treatments.

**Source data 2.** Original files for western blot analysis displayed in *Figure 8D*.

**Figure supplement 1.** STAMBPL1, FOXO1, and GRHL3 is highly expressed in triple-negative breast cancer (TNBC).

been elucidated. The transcriptional activity of FOXO1 is primarily regulated by its nucleocytoplasmic shuttling process (*Van Der Heide et al., 2004*). The PI3K/AKT pathway promotes the phosphorylation of FOXO1, resulting in the formation of a complex with members of the 14-3-3 family (including 14-3-3σ, 14-3-3ε, and 14-3-3 ζ ), which facilitates its export from the nucleus and inhibits its transcriptional activity (*Huang and Tindall, 2007*; *Tzivion et al., 2011*). It is reported that TDAG51 prevents the binding of 14-3-3 ζ to FOXO1 in the nucleus by interacting with FOXO1, thereby enhancing its transcriptional activity through increased accumulation within the nucleus (*Park et al., 2023*). Our results indicate that the overexpression of STAMBPL1 and STAMBPL1-E292A did not affect the protein levels of FOXO1 (*Figure 7E*, *Figure 7—figure supplement 1E*), but STAMBPL1 co-localizes with FOXO1 in the nucleus (*Figure 7M*) and interacts with it (*Figure 7N*, *Figure 7—figure supplement 1I,J*). This suggests that STAMBPL1 enhances the transcriptional activity of FOXO1 on GRHL3 by interacting with nuclear FOXO1. So, it will be important to develop a *Stambpl1* knockout mouse model to investigate the exact role of STAMBPL1 in TNBC. Additionally, we need to develop HIF1α and FOXO1 antibodies suitable for immunohistochemistry to detect their expression in TNBC clinical samples.

Several studies have reported that FOXO1 inhibits tumor angiogenesis (*Wei et al., 2022*; *Kim et al., 2016*; *Dai et al., 2022*; *Shang et al., 2020*; *Liu et al., 2024*). Studies have shown that M2 macrophage-derived exosomal miR-942 promotes the migration and invasion of lung adenocarcinoma cells and facilitates angiogenesis by binding to FOXO1 to alleviate the inhibition of β-catenin, in which the upregulation of FOXO1 induces a decrease in cell invasion and angiogenesis in vitro (*Wei et al., 2022*). FOXO1 inhibits gastric cancer growth and angiogenesis under hypoxic conditions via inactivation of the HIF1α-VEGF pathway, possibly in association with SIRT1 (*Kim et al., 2016*). Cancer-associated fibroblast (CAF)-derived extracellular vesicles deliver miR-135b-5p into colorectal adenocarcinoma cells to downregulate FOXO1 and promote HUVEC proliferation, migration, and angiogenesis (*Dai et al., 2022*). Colorectal cancer cell-derived exosomes overexpressing miR-183-5p promote the proliferation, migration and tube formation of HMEC-1 (human microvascular endothelial cells) cells through the inhibition of FOXO1 (*Shang et al., 2020*). Bladder cancer cell-derived exosomal miR-1247-3p facilitates angiogenesis by inhibiting FOXO1 expression (*Liu et al., 2024*). However, the role of FOXO1 in breast cancer angiogenesis has not been studied, and our study revealed that FOXO1 can promote the expression of HIF1α and VEGFA, suggesting that it may play a role in promoting angiogenesis in breast cancer.

Studies have demonstrated that the AKT-FOXO1 signaling pathway regulates the expression of GRHL3. To investigate this, the researchers utilized a previously published ChIP-seq dataset for FOXO1 from human endometrial stromal cells. They reported that FOXO1 occupied BRD4-bound enhancers near the *GRHL3* gene. The authors subsequently confirmed that the removal of EGF and insulin from the growth medium significantly increased the expression of GRHL3, but the administration of the FOXO1 inhibitor AS1842856 partially blocked the induced expression of GRHL3 (*Nagarajan et al., 2017*). These results suggest that FOXO1 plays a regulatory role upstream of GRHL3, and our study also confirmed that FOXO1 promotes the transcription of *GRHL3* by binding to its promoter.

GRHL3 is a highly conserved epidermal-specific developmental transcription factor that has recently gained attention in the field of cancer research (*Darido and Jane, 2010*). To date, only a few genes regulated by GRHL3 have been identified. Studies have shown that GRHL3 levels are significantly reduced in human skin and head and neck squamous cell carcinomas and suggest that GRHL3 is a key tumor suppressor pathway in squamous cell carcinomas (*Darido et al., 2011*). GRHL3 was also found to be induced by TNF-α in the mammary carcinoma cell line MCF-7 and was identified as a TNFα-induced endothelial cell migration factor with promigratory activity as high as that of VEGF (*Guardiola-Serrano et al., 2008*). However, the specific role of GRHL3 in breast cancer is unclear.

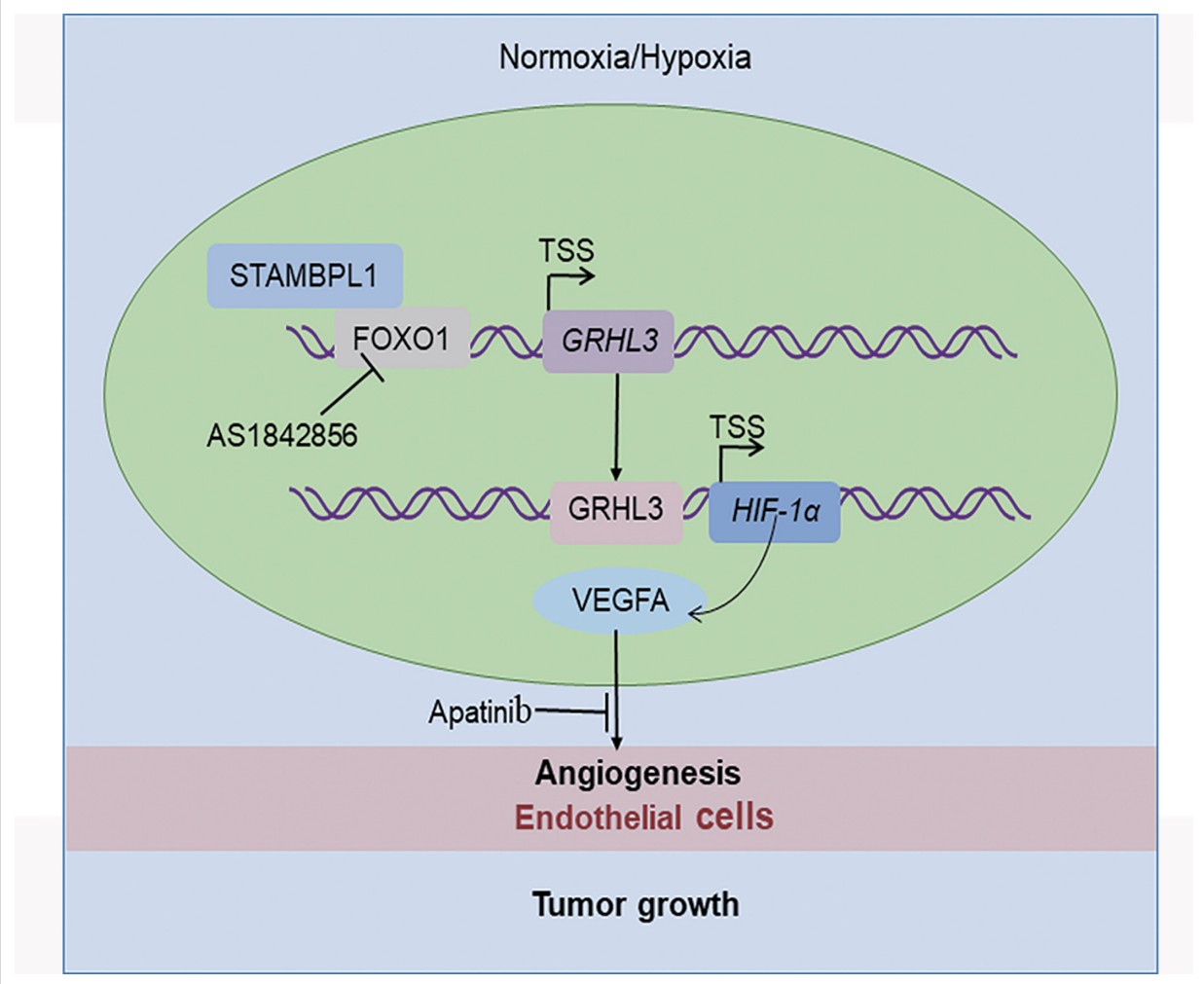

**Figure 9.** The working model of this study. STAMBPL1 interacts with FOXO1 to promote triple-negative breast cancer (TNBC) angiogenesis by activating the GRHL3/HIF1α/VEGFA axis.

Studies have confirmed that GRHL3 strongly stimulates endothelial cell migration, which is consistent with an angiogenic, protumorigenic function (*Guardiola-Serrano et al., 2008*). In our study, we demonstrated that GRHL3 promotes TNBC angiogenesis. HIF1α, a known regulator of angiogenesis, is regulated by growth factors (*Fukuda et al., 2002*; *Zhong et al., 2000*), cytokines (*Hellwig-Bürgel et al., 2005*), and mitogens (*Kasuno et al., 2004*; *Li et al., 2004*). The EGFR/Akt pathway is a known positive regulator of HIF1α, potentially through mTOR (*Fukuda et al., 2002*) or independent of it (*Mazure et al., 1997*; *Pore et al., 2006*). While the mechanisms regulating HIF1α protein expression have been extensively studied, those modulating HIF1α transcriptional activity remain unclear. Our study reveals for the first time that GRHL3 promotes transcriptional activity by binding to the promoter of the *HIF1A* gene.

In this study, we utilized two TNBC cell lines, HCC1806 and HCC1937, along with human primary umbilical vein endothelial cells (HUVECs) and a nude mouse breast orthotopic transplantation tumor model to investigate the regulatory mechanism by which STAMBPL1 activates the GRHL3/HIF1α/VEGFA signaling pathway through its interaction with FOXO1, thereby promoting angiogenesis in TNBC. The results of this study have certain limitations regarding their applicability to human TNBC biology. Furthermore, in addition to the HIF1α/VEGFA signaling pathway emphasized in this study, tumor cells can continuously release or upregulate various pro-angiogenic factors, such as Angiopoietin and FGF, which activate endothelial cells, pericytes, CAFs, endothelial progenitor cells, and immune cells. This leads to capillary dilation, basement membrane disruption, extracellular matrix remodeling, pericyte detachment, and endothelial cell differentiation, thereby sustaining a highly

active state of angiogenesis (*Liu et al., 2023*). It is important to collect clinical TNBC tissue samples in the future to analyze the expression of the STAMBPL1/FOXO1/GRHL3/HIF1α/VEGFA signaling axis. Furthermore, patient-derived organoid and xenograft models are useful to elucidate the regulatory relationship of this axis in TNBC angiogenesis.

## Conclusions

In summary, STAMBPL1 promoted TNBC angiogenesis by activating the GRHL3/HIF1α/VEGFA pathway via interacting with FOXO1 in a non-enzymatic manner. These findings highlight the significant role of STAMBPL1 in TNBC angiogenesis and suggest that targeting the STAMBL1/FOXO1/GRHL3/HIF1α/VEGFA axis could be a potential therapeutic strategy to inhibit angiogenesis in TNBC.

## Acknowledgements

We sincerely thank the team members for their dedication to this study. This work was supported by National Key Research and Development Program of China (2023YFA1800500, 2023ZD0502200, 2020YFA0112300), National Natural Science Foundation of China (U2102203 and 82430084 to CC, 82203413 to HL), Biomedical Projects of Yunnan Key Science and Technology Program (202302AA310046 to CC), Yunnan Fundamental Research Projects (202201BC070002 and 202301AS070050 to CC, 202201AT070290 to HL), Yunnan Revitalization Talent Support Program (Yunling Shcolar Project to CC), Yunnan (Kunming) Academician Expert Workstation (grant No. Q7 YSZJGZZ-2020025 to CC), The Innovative Research Team of Yunnan Province (202405AS350016 to CC), Kunming University of Science and Technology-The First People's Hospital of Yunnan Province Joint Major Project (No. KUST-KH2022005Z), and Center of Clinical Pharmacy of the First People's Hospital of Yunnan Province Open Project (No. 2023YJZX-YX04).

## Additional information

### Funding

| Funder | Grant reference number | Author |
| --- | --- | --- |
| National Key Research and Development Program of China | 2023YFA1800500 | Huichun Liang |
| National Key Research and Development Program of China | 2023ZD0502200, 2020YFA0112300 | Ceshi Chen |
| National Natural Science Foundation of China | U2102203, 82430084 | Ceshi Chen |
| National Natural Science Foundation of China | 82203413 | Huichun Liang |
| Biomedical Projects of Yunnan Key Science and Technology Program | 202302AA310046 | Ceshi Chen |
| Yunnan Fundamental Research Projects | 202301AS070050, 202201BC070002 | Ceshi Chen |
| Yunnan Fundamental Research Projects | 202201AT070290 | Huichun Liang |
| Yunnan Revitalization Talent Support Program | Yunling Shcolar | Ceshi Chen |
| Yunnan (Kunming) Academician Expert Workstation | No. Q7 YSZJGZZ-2020025 | Ceshi Chen |

| Funder | Grant reference number | Author |
| --- | --- | --- |
| Kunming University of Science and Technology-The First People's Hospital of Yunnan Province Joint Major Project | KUST-KH2022005Z | Huifeng Zhang |
| Center of Clinical Pharmacy of the First People's Hospital of Yunnan Province Open Project | 2023YJZX-YX04 | Huifeng Zhang |
| The Innovative Research Team of Yunnan Province | 202405AS350016 | Ceshi Chen |

The funders had no role in study design, data collection, and interpretation, or the decision to submit the work for publication.

## Author contributions

Huan Fang, Data curation, Formal analysis, Investigation, Visualization, Writing – original draft; Huichun Liang, Conceptualization, Data curation, Formal analysis, Supervision, Funding acquisition, Validation, Visualization, Methodology, Writing – original draft, Writing – review and editing; Chuanyu Yang, Project administration; Dewei Jiang, Supervision, Project administration; Qianmei Luo, Investigation; Wen-Ming Cao, Resources, Funding acquisition; Huifeng Zhang, Supervision, Funding acquisition; Ceshi Chen, Conceptualization, Data curation, Supervision, Funding acquisition, Validation, Project administration, Writing – review and editing

## Author ORCIDs

Huan Fang http://orcid.org/0009-0003-0489-5796
Huichun Liang https://orcid.org/0000-0002-9157-3087
Ceshi Chen https://orcid.org/0000-0001-6398-3516

## Ethics

We purchased 5- to 6-week-old female BALB/c nude mice from SLACCAS (Changsha, China). Animal feeding and experiments were approved by the animal ethics committee of Kunming Institute of Zoology, Chinese Academy of Sciences (IACUC-PA-2022-03-029).

Reviewer #1 (Public review): https://doi.org/10.7554/eLife.102433.3.sa1
Reviewer #2 (Public review): https://doi.org/10.7554/eLife.102433.3.sa2
Reviewer #3 (Public review): https://doi.org/10.7554/eLife.102433.3.sa3
Author response https://doi.org/10.7554/eLife.102433.3.sa4

# Additional files

## Supplementary files

MDAR checklist

## Data availability

All data generated or analyzed during this study are included in the manuscript and supporting files; source data files have been provided for Figures 1, 2, and 4–8.

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
