## [Editor Report · eLife Assessment]

The study conducted by Fang et al. offers significant and **fundamental** insights, notably enhancing our understanding of angiogenesis. While some of the claims are supported by **convincing** experimental approaches, others lack sufficient validation. Additionally, there are instances where critical experimental controls appear to be absent.

---

## [Referee Report · Reviewer #1 (Public review)]

Summary:

In this study by Fang et al., the authors show how STAMBPL1 promotes TNBC angiogenesis via a feed-forward GRHL3/HIF1a/VEGFA axis. They demonstrate that STAMBPL1 interacts with FOXO1, define the required domains in each protein, and illustrate that this interaction facilitates FOXO1 transcriptional factor activity, which then activates GRHL3/HIF1a/VEGFA signaling. Lastly, they show that the combination of VEGFR and FOXO1 inhibitors can synergistically suppress STAMBPL1-overexpressing TNBC.

Strengths:

The manuscript is clearly written, and the results are well explained. The observation that STAMBPL1 mediates GRHL3 transcription through its interaction with FOXO1 is novel. The findings also have important translational potential.

---

## [Referee Report · Reviewer #2 (Public review)]

Summary:

In their manuscript, Fang and colleagues make a notable contribution to the field of oncology, particularly in advancing our understanding of triple-negative breast cancer (TNBC). The research delineates the role of STAMBPL1 in promoting angiogenesis in TNBC through its interaction with FOXO1 and the subsequent activation of the GRHL3/HIF1A/VEGFA axis. The evidence presented is robust, with a combination of in vitro experiments, RNA sequencing, and in vivo studies providing a comprehensive view of the molecular mechanisms at play. The strength of the evidence is anchored in the systematic approach and the utilization of multiple methodologies to substantiate the findings.

Strengths:

The manuscript presents a methodologically robust framework, incorporating RNA-sequencing, chromatin immunoprecipitation (ChIP) assays, and a suite of in vitro and in vivo model systems, which collectively substantiate the claims regarding the pro-angiogenic role of STAMBPL1 in TNBC. The employment of multiple cellular models, conditioned media to assess HUVEC functional responses, and xenograft tumor models in murine hosts offers a comprehensive evaluation of STAMBPL1's impact on angiogenic processes.A salient strength of this work is the identification of GRHL3 as a transcriptional target of STAMBPL1 and the demonstration of a physical interaction between STAMBPL1 and FOXO1, which modulates GRHL3-driven HIF1A transcription. The study further suggests a potential therapeutic strategy by revealing the synergistic inhibitory effects of combined VEGFR and FOXO1 inhibitor treatment on TNBC tumor growth.

Weaknesses:

A potential limitation of the study is the reliance on specific cellular and animal models, which may constrain the extrapolation of these findings to the broader spectrum of human TNBC biology. Furthermore, while the study provides evidence for a novel regulatory axis involving STAMBPL1, FOXO1, and GRHL3, the multifaceted nature of angiogenesis may implicate additional regulatory factors not exhaustively addressed in this research.

Appraisal of Achievement and Conclusion Support:

The authors have successfully demonstrated that STAMBPL1 promotes HIF1A transcription and activates the HIF1α/VEGFA axis in a non-enzymatic manner, leading to increased angiogenesis in TNBC. The results are generally supportive of their conclusions, with clear evidence that STAMBPL1 upregulates HIF1α expression and enhances the activity of HUVECs. The study also shows that STAMBPL1 interacts with FOXO1 to promote GRHL3 transcription, which in turn activates HIF1A.

Impact on the Field and Utility:

This research is poised to exert a substantial impact on the oncological research community by uncovering the role of STAMBPL1 in TNBC angiogenesis and by identifying the STAMBPL1/FOXO1/GRHL3/HIF1α/VEGFA axis as a potential therapeutic target. The findings could pave the way for the development of novel therapeutic strategies for TNBC, a subtype characterized by a paucity of effective treatment options. The methodologies utilized in this study are likely to be valuable to the research community, offering a paradigm for investigating the role of deubiquitinating enzymes in oncogenic processes.

Additional Context:

It would be beneficial for readers to understand the broader context of TNBC research and the current challenges in treating this aggressive cancer subtype. The significance of this work is heightened by the lack of effective treatments for TNBC, making the identification of new therapeutic targets particularly important. Furthermore, understanding the specific mechanisms by which STAMBPL1 regulates HIF1α expression could provide insights into hypoxia signaling in other cancer types as well.

---

## [Referee Report · Reviewer #3 (Public review)]

In this manuscript, Fang et al. describe a new oncogenic function of the STAMBPL1 protein in triple-negative breast cancer (TNBC). STAMBPL1 is a deubiquitinase that has been poorly studied in cancer. Previous reports identify it as a promoter of epithelial to mesenchymal transition or an inhibitor of cisplatin-induced cell death, but its participation to other cancer phenotypes has not been investigated. Fang et al. find that in cell line models of TNBC, STAMBPL1 promotes expression of the transcription factor HIF-1a and its downstream target VEGF, with the consequence of stimulating neo-angiogenesis in vitro and in vivo. Mechanistically, the authors find that this occurs via a non-enzymatic and indirect mechanism, that is by promoting the expression of GRHL3, a transcription factor that in turn binds to the HIF-1a promoter to stimulate its transcription. Interestingly, the way by which STAMPB1 promotes GRHL3 expression is by facilitating the transcriptional activity of FOXO1, a known regulator of GRHL3. Because the authors find that STAMBPL1 and FOXO1 interact, they suggest that STAMBPL1 may promote the formation of an active transcriptional complex containing FOXO1, perhaps by facilitating the recruitment of transcriptional coactivators.

In conclusion, these data position for the first time the STAMBPL1 deubiquitinase in a FOXO-GRHL3 regulatory axis for the control of VEGF expression and tumor angiogenesis.

The main weaknesses of this work are that the relevance of this molecular axis to the pathogenesis of TNBC is not clear, and it is not clearly established whether this is a regulatory pathway that occurs in hypoxic conditions or independently of oxygen levels.

Major criticisms:

(1) Both FOXO1 and GRHL3 have been previously described as tumor suppressors, with reports of FOXO1 inhibiting tumor angiogenesis. Therefore, this work describes an apparently contradictory function of these proteins in TNBC. While it is not surprising that the same genes perform divergent functions in different tumor contexts, a stronger evidence in support of the oncogenic function of these two genes should be provided to make the data more convincing.

To strengthen the notion that STAMBPL1, FOXO and GRHL3 are overexpressed in TNBC, the authors have utilized the BCIP tool to analyze their expression in the Metabric database. According to this analysis, the levels of STAMBPL1and GRHL3 are not higher in breast cancer than in adjacent tissues, and the levels of FOXO1 are lower. Nonetheless, the authors observe that their expression levels are significantly (yet not dramatically) higher in TNBC compared to non-TNBC (Fig.S6A-C). However, these new data do not provide convincing evidence of the relevant tumor suppressive function of these genes in TNBC, as neither is more expressed in tumors compared to adjacent normal tissues.

(2) Because STAMBPL1 overexpression in normoxic conditions is sufficient to cause HIF-1a protein accumulation, it is not clear why the authors then use hypoxic conditions to analyze the effect of STAMBPL1 on HIF-1a transcription Avoiding HIF1-a protein degradation should not have any effect on its transcription. At the same time, it is not clear nor is being explained why different hypoxic conditions are sometimes used, resulting in different mRNA levels of HIF-1a and its downstream targets and quite significant fluctuations within the same cell line from one experimental setting to the next. In conclusion, it is not clear what is the relevance of the new HIF-1a regulatory axis described in this paper in normoxic or hypoxic conditions.

(3) Another critical point is that necessary experimental controls are sometimes missing, and this is reducing the strength of some of the conclusions enunciated by the authors. As an example, experiments where overexpression of STAMBPL1 is coupled to silencing of FOXO1 to demonstrate dependency lack FOXO1silencing the absence of STAMBPL1 overexpression. Because diminishing FOXO1 expression affects HIF-1a/VEGF transcription even in the absence of STAMBPL1 (shown in Figure 7C, D), it is not clear if the data presented in Figure 7G are significant. The difference between HIF-1a expression upon FOXO1 silencing should be compared in the presence or absence of STAMBPL1 overexpression to understand if FOXO1 impacts HIF-1a transcription dependently or independently of STAMBPL1.

In addition, some minor comments to improve the quality of this manuscript are provided.

(1) In Figures 2A and D, where endogenous versus STAMBPL1 expression is shown, it is not clear what is the molecular weight of these proteins as they both appear to be of 55 KDa, even though according to the authors the exogenous protein is bigger than the endogenous and the lower band in Figure 2D is reported to be the endogenous STAMBPL1.

(2) In Figure 2, the effect of STAMBPL1 overexpression on HIF-1a mRNA is minor. At the same time, it seems that the protein levels of HIF-1a are quite high (or at least visible by WB) in normoxic cells even in the absence of STAMBPL1 overexpression. This raises questions about the type of regulation that HIF-1a is subjected to in these cells.

In general, because only two cell lines are used in this study and the data in patients do not appear to strongly support an oncogenic function of STAMBPL1 in TNBC (via its overexpression), data should be more solid and additional experiments should be provided to substantiate the oncogenic function of this pathway in TNCB.

---

## [Author Response]

The following is the authors’ response to the original reviews.

**Reviewer #1 (Public review):**
(1) The mechanism by which STAMBPL1 mediates GRHL3 transcription through its interaction with FOXO1 is not sufficiently discussed, especially in relation to how STAMBPL1 regulates FOXO1. Some reported effects are modest.

We appreciate the reviewer’s comments. In response, we have added a discussion on the potential mechanisms by which STAMPBL1 regulates FOXO1 transcriptional activity in Discussion, highlighted in red on page 18, lines 342 to 352. The specific reply content is as follows: “The transcriptional activity of FOXO1 is primarily regulated by its nucleocytoplasmic shuttling process (Van Der Heide, Hoekman et al. 2004). The PI3K/AKT pathway promotes the phosphorylation of FOXO1, resulting in the formation of a complex with members of the 14-3-3 family (including 14-3-3σ, 14-3-3ε, and 14-3-3ζ), which facilitates its export from the nucleus and inhibits its transcriptional activity (Huang and Tindall 2007, Tzivion, Dobson et al. 2011). It’s reported that TDAG51 prevents the binding of 14-3-3ζ to FOXO1 in the nucleus by interacting with FOXO1, thereby enhancing its transcriptional activity through increased accumulation within the nucleus (Park, Jeon et al. 2023). Our results indicate that the overexpression of STAMBPL1 and STAMBPL1-E292A did not affect the protein levels of FOXO1 (Fig.7E and Fig.S5E), but STAMBPL1 co-localizes with FOXO1 in the nucleus (Fig.7M) and interacts with it (Fig.7N and Fig.S5I-J). This suggests that STAMBPL1 enhances the transcriptional activity of FOXO1 on GRHL3 by interacting with nuclear FOXO1.” The result was added to Supplementary Figure 5 as Fig.S5E.

**Reviewer #2 (Public review):**
(1) A potential limitation of the study is the reliance on specific cellular and animal models, which may constrain the extrapolation of these findings to the broader spectrum of human TNBC biology. Furthermore, while the study provides evidence for a novel regulatory axis involving STAMBPL1, FOXO1, and GRHL3, the multifaceted nature of angiogenesis may implicate additional regulatory factors not exhaustively addressed in this research.

We appreciate the valuable suggestions provided by the reviewer. In Discussion, we have added an in-depth discussion of the limitations of the study, as well as an analysis of the regulatory factors related to tumor angiogenesis, which highlighted in red on pages 20 to 21, lines 396 to 412. The relevant content added is as follows: “In this study, we utilized two triple-negative breast cancer cell lines, HCC1806 and HCC1937, along with human primary umbilical vein endothelial cells (HUVECs) and a nude mouse breast orthotopic transplantation tumor model to investigate the regulatory mechanism by which STAMBPL1 activates the GRHL3/HIF1α/VEGFA signaling pathway through its interaction with FOXO1, thereby promoting angiogenesis in TNBC. The results of this study have certain limitations regarding their applicability to human TNBC biology. Furthermore, in addition to the HIF1α/VEGFA signaling pathway emphasized in this study, tumor cells can continuously release or upregulate various pro-angiogenic factors, such as Angiopoietin and FGF, which activate endothelial cells, pericytes (PCs), cancer-associated fibroblasts (CAFs), endothelial progenitor cells (EPCs), and immune cells (ICs). This leads to capillary dilation, basement membrane disruption, extracellular matrix remodeling, pericyte detachment, and endothelial cell differentiation, thereby sustaining a highly active state of angiogenesis (Liu, Chen et al. 2023). It is important to collect clinical TNBC tissue samples in the future to analyze the expression of the STAMBPL1/FOXO1/GRHL3/HIF1α/VEGFA signaling axis. Furthermore, patient-derived organoid and xenograft models are useful to elucidate the regulatory relationship of this axis in TNBC angiogenesis”

**Reviewer #3 (Public review):**
The main weaknesses of this work are that the relevance of this molecular axis to the pathogenesis of TNBC is not clear, and it is not clearly established whether this is a regulatory pathway that occurs in hypoxic conditions or independently of oxygen levels.(1) With respect to the first point, both FOXO1 and GRHL3 have been previously described as tumor suppressors, with reports of FOXO1 inhibiting tumor angiogenesis. Therefore, this works describes an apparently contradictory function of these proteins in TNBC. While it is not surprising that the same genes perform divergent functions in different tumor contexts, a stronger evidence in support of the oncogenic function of these two genes should be provided to make the data more convincing. As an example, the data in support of high STAMBPL1, FOXO and GRHL3 gene expression in TNBC TCGA specimens provided in Figure 8 is not very strong and it is not clear what the non-TNBC specimens are (whether other breast cancers or other tumors, perhaps those tumors whether these genes perform tumor suppressive functions). To strengthen the notion that STAMBPL1, FOXO and GRHL3 are overexpressed in TNCB, the authors could provide a comparison with normal tissue, as well as the analysis of other publicly available datasets (like the NCI Clinical Proteomic Tumor Analysis Consortium as an example). Finally, is it not clear what are the basal protein expression levels of STAMBPL1 in the cell lines used in this study, as based on the data presented in Figures 2D and F it appears that the protein is not expressed if not exogenously overexpressed. It would be helpful if the authors addressed this issue and provided further evidence of STAMBPL1 expression in TNBC cell lines.

We appreciate the suggestions. In this study, we utilized the BCIP online tool to analyze the Metabric database, incorporating adjacent normal tissues as controls. Although the expression levels of STAMBPL1, FOXO1, and GRHL3 in breast cancer tissues are not uniformly higher than those in adjacent tissues, their expression levels in triple-negative breast cancer (TNBC) are significantly elevated compared to non-TNBC. The results of this re-analysis have been added in Supplementary Figure 6 as Fig.S6A-C.

About the question of the basal protein expression levels of STAMBPL1 in the cell lines used in this study, our response is that Fig. 2A showed the endogenous level of STAMBPL1 in HCC1806 and HCC1937. For Fig. 2D and 2F, the overexpressed STAMBPL1 was fused with a 3xFlag tag, resulting in a higher molecular weight compared to the endogenous STAMBPL1. In the revised Figure 2, we have indicated the positions of the endogenous (Endo.) and exogenous (OE.) STAMBPL1 bands with arrows.

(2) Linked to these considerations is the second major criticism, namely that it is not made clear if this new regulatory axis is proposed to act in normoxic or hypoxic conditions. The experiments presented in this paper are performed in both conditions but a clear explanation as to why cells are exposed to hypoxia is not given and would be necessary being that HIF-1a transcription and not protein stability is being analyzed. Also, different hypoxic conditions are sometimes used, resulting in different mRNA levels of HIF-1a and its downstream targets and quite significant fluctuations within the same cell line from one experimental setting to the next. The authors should provide an explanation as to why experimental conditions are changed and, more importantly, the experiments presented in Figure 2 should be performed also in normoxia.

Thanks for the comments. Under normoxic conditions, HIF1α is recognized by pVHL due to hydroxylation and is rapidly degraded via the proteasomal pathway. In contrast, under hypoxic conditions, HIF1α protein is accumulated. To investigate the effect of STAMBPL1 knockdown on *HIF1A* gene transcription levels, we conducted experiments under hypoxic conditions to avoid interference from the rapid degradation of HIF1α at the protein level, as shown in Figures 2B-C. Furthermore, under normoxic conditions, the overexpression of STAMBPL1 had been demonstrated to significantly enhance the protein levels of HIF1α and upregulate the transcription of *VEGFA* through HIF1α. To avoid the potential impact of excessive accumulation of HIF1α protein under hypoxic conditions on its protein level detection and the transcription of downstream VEGFA, the related experiments shown in Figure 2D-G were performed under normoxic conditions. We have explained the corresponding experimental conditions in the “Result” and “Figure legends” according to the reviewer's comments, highlighted in red.

(3) Another critical point is that necessary experimental controls are sometimes missing, and this is reducing the strength of some of the conclusions enunciated by the authors. As examples, experiments where overexpression of STAMBPL1 is coupled to silencing of FOXO1 to demonstrate dependency lack FOXO1 silencing the absence of STAMBPL1 overexpression. Because diminishing FOXO1 expression affects HIF-1a/VEGF transcription even in the absence of STAMBPL1 (shown in Figure 7C, D), it is not clear if the data presented in Figure 7G are significant. The difference between HIF-1a expression upon FOXO1 silencing should be compared in the presence or absence of STAMBPL1 overexpression to understand if FOXO1 impacts HIF-1a transcription dependently or independently of STAMBPL1.

Thank you for this comment. For Fig.7G-H, our experimental objective was to determine whether the activation of *HIF1A*/*VEGFA* transcription by STAMBPL1 via FOXO1. Therefore, under STAMBPL1 overexpression, we knocked down FOXO1 to investigate whether FOXO1 silencing could reverse the upregulation of *HIF1A*/*VEGFA* transcription induced by STAMBPL1 overexpression.

(4) In addition, some minor comments to improve the quality of this manuscript are provided.(4.1) As a general statement, the manuscript is extremely synthetic. While this is not necessarily a negative feature, sometimes results are discussed in the figure legends and not in the main text (as an example, western blots showing HIF-1a expression) and this makes it hard to read thought the data in an easy and enjoyable manner.

Thank you for this suggestion. We have revised the figure legends to make them clearer and more concise, highlighted in red.

(4.2) The effect of STAMBPL1 overexpression on HIF-1a transcription is minor (Figure 2) The authors should explain why they think this is the case and whether hypoxia may provide a molecular environment that is more permissive to this type of regulation.

Thank you for the comment. Under normoxic conditions, we conducted WB to examine the protein expression of HIF1α after the overexpression of STAMBPL1 and the knockdown of HIF1α. To visually illustrate the impact of STAMBPL1 overexpression on HIF1A protein levels, as well as the effectiveness of HIF1α knockdown, we annotated the grayscale analysis results of the bands in Figures 2D and 2F. As the reviewer pointed out, under normoxic conditions, HIF1α is rapidly degraded, which may explain why the upregulation of HIF1α protein levels by STAMBPL1 overexpression is not very pronounced.

(4.3) HIF-1a does not appear upregulated at the protein level protein by STAMBPL1 or GRLH3 overexpression, even though this is stated in the legends of Figures 2 and 6. The authors should show unsaturated western blots images and provide quantitative data of independent experiments to make this point.

Thank you for this comment. We have added the unsaturated image of HIF1α into Fig.2D, and performed a grayscale analysis of the HIF1α bands in Fig.2D and Fig.6A to indicate the relative protein level of HIF1α.

**Reviewer #1 (Recommendations for the authors):**
(1) The authors previously reported that STAMBPL1 stabilizes MKP1 in TNBC. However, in this study, they focus on HIF1a. Given that STAMBPL1 affects HIF1a expression, it would be valuable to examine the levels of ROS in TNBC cells with or without STAMBPL1, as ROS is known to influence HIF1a stability.

Thank you for your comments. It’s known that STAMBPL1 functions as a deubiquitinating enzyme. However, our study reveals that the upregulation of HIF1α by STAMBPL1 is independent of its deubiquitinating activity. This conclusion is supported by the observation that overexpression of the deubiquitinase active site mutant, STAMBPL1-E292A, also upregulated HIF1α expression (Figure 1F). Moreover, STAMBPL1 overexpression enhanced HIF1α transcription (Figures 4E and S3E), while STAMBPL1 knockdown was able to inhibit the transcription of HIF1α (Figures 2B-C). These results indicate that STAMBPL1 mediates the transcription of HIF1α but does not affect the stability of HIF1α. For these reasons, we think that it is unnecessary to examine the ROS levels.

(2) Figure 1A: The regulation of HIF1a mRNA by STAMBPL1, but not its protein levels, could be better addressed by using MG132 to rule out the impact of protein degradation.

Thanks for this comment. Under normoxic conditions, the oxygen-sensitive prolyl hydroxylases PHD1-3 act on HIF1α, specifically inducing hydroxylation at the proline 402 and 564 residues. These hydroxylated residues are recognized by the pVHL/E3 ubiquitin ligase complex, leading to ubiquitination and subsequent degradation via the proteasome pathway. Conversely, under hypoxic conditions, PHD1-3 are inactivated, and non-hydroxylated HIF1α is not recognized by the pVHL/E3 ubiquitin ligase complex, thereby avoiding ubiquitination and proteasomal degradation (DOI: 10.1073/pnas.95.14.7987, DOI: 10.1515/BC.2004.016, and DOI: 10.1042/BJ20040620). The mechanism of HIF1α accumulation under hypoxia is analogous to the action of the proteasome inhibitor MG132. When we treated cells with hypoxia, the ubiquitination and proteasomal degradation pathway of HIF1α was blocked. At this time, STAMBPL1 knockdown could downregulate the expression of HIF1α (Fig.1A). Meanwhile, since the knockdown of STAMBPL1 significantly downregulated the mRNA level of HIF1α under hypoxia (Fig.2B-C), we concluded that STAMBPL1 affects the expression of HIF1α by mediating its transcription. In addition, MG132 will block all proteasomal substrate degradation and may affect HIF1α mRNA levels indirectly.

(3) Figure 2D and 2F: The effect of STAMBPL1 in promoting HIF1a expression is quite mild, and the effect of HIF1a knockdown is also modest. Given the high levels of STAMBPL1 in TNBC cell lines (Figure 2A), it would be better to repeat these experiments in a STAMBPL1-knockdown setting for clearer insights.

We appreciate this insightful suggestion. Considering that the regulation of HIF1α expression by STAMBPL1 occurs at the transcriptional level, and to prevent excessive accumulation of HIF1a during hypoxia that could confound the effect of STAMBPL1 overexpression on HIF1α regulation, we opted to overexpress STAMBPL1 under normoxic conditions and subsequently knock down HIF1α, as shown in Fig.2D and Fig.2F. This approach allowed us to observe that STAMBPL1 overexpression can upregulate HIF1a expression to some extent. Additionally, in response to the reviewer's suggestion to knock down STAMBPL1, we have conducted the corresponding experiments, with results presented in Fig.1A-E and Fig.2B-C.

(4) Figure 4A: Why does the RNA-seq pattern differ significantly between the two siRNAs? Additionally, the authors should clarify why they focus primarily on transcription factors, as other mechanisms, such as mRNA stability and RNA modification, could also influence gene transcription.

Thank you for this comment. Two siRNAs for STAMBPL1 were designed and synthesized by a biotechnology company. Although both siRNAs target STAMBPL1, they target different sequences. While both siRNAs effectively knocked down STAMBPL1 (Fig. 1A and Fig. 2A), the possibility of off-target effects cannot be completely ruled out. Therefore, we needed to use two siRNAs simultaneously for RNA-seq, ensuring that the gene expression changes observed are due to the knockdown of STAMBPL1 by focusing on genes downregulated by both two siRNAs. Additionally, among the 27 genes downregulated by both two siRNAs, only 18 genes were annotated. Of these 18 genes, except for GRHL3, which is a transcription factor reported to be involved in gene transcription regulation, the remaining 17 genes have no documented association with RNA transcription, stability, or modification. Therefore, we focused on the GRHL3 gene.

(5) Figure 5G: To investigate whether STAMBPL1 and GRHL3 function epistatically in the pathway, a double knockdown of STAMBPL1 and GRHL3 should be examined. Additionally, a double knockdown of STAMBPL1 and FOXO1 should be assessed.

Thank you for your comment. In Figure 5G, we aimed to assess the knockdown efficiency of GRHL3 using siRNAs. To determine whether STAMBPL1 upregulates the HIF1a/VEGFA axis via GRHL3, we overexpressed STAMBPL1 and subsequently knocked down GRHL3. Our findings indicated that STAMBPL1 overexpression indeed enhanced the HIF1a/VEGFA axis, which was rescued by the knockdown of GRHL3, as shown in Figures 4E-F and S3E-F. Similarly, upon overexpressing STAMBPL1 and knocking down FOXO1, we observed that STAMBPL1 overexpression increased the GRHL3/HIF1a/VEGFA axis, which could also be rescued by knocking down FOXO1, as shown in Figures 7F-H. These results suggest that STAMBPL1 upregulates the GRHL3/HIF1a/VEGFA axis through FOXO1. We do not think it is a right way to double knock down STAMBPL1 and FOXO1 or GRHL3.

(6) Figure 7: It remains unclear how STAMBPL1 regulates FOXO1. The authors show that STAMBPL1 increases the transcriptional activation of FOXO1 at the GRHL3 promoter, but it is not clear if STAMBPL1 is required for FOXO1 binding to the GRHL3 promoter. To address this, STAMBPL1-knockdown should be included to examine its effect on FOXO1 binding to the GRHL3 promoter. Furthermore, it would be important to determine whether the STAMBPL1-FOXO1 interaction is essential for GRHL3 transcription. Since the interaction sites of STAMBPL1-FOXO1 have been mapped, a mutant disrupting the interaction would provide better insight into how STAMBPL1 promotes GRHL3 transcription by interacting with FOXO1.

Thank you for this comment. It has been reported that FOXO1 promotes the transcription of the *GRHL3* gene by interacting with its promoter (DOI: 10.1093/nar/gkw1276). We also verified through ChIP assay that FOXO1 can bind to the promoter of *GRHL3* gene (Fig.7I) and mediate its transcription. Specifically, knocking down FOXO1 significantly down-regulated the mRNA level of GRHL3 (Fig.7B), and the *GRHL3* promoter lacking FOXO1 binding site almost completely lost transcriptional activity (Fig.7J), indicating that FOXO1 is crucial for the transcriptional activity of the *GRHL3* promoter. Overexpression of STAMBPL1 enhances the activating effect of FOXO1 on the transcriptional activity of the *GRHL3* promoter (Fig.7K). However, the up-regulation of GRHL3 transcription by overexpression of STAMBPL1 is completely blocked by FOXO1 knockdown (Fig.7F), and the knockdown of FOXO1 essentially blocks the binding of STAMBPL1 to the *GRHL3* promoter (Fig.7L), suggesting that STAMBPL1 affects the transcriptional expression of GRHL3 based on FOXO1. As we added in Discussion, the transcription factor activity of FOXO1 is mainly regulated by its nucleoplasm shuttling process, and the accumulation of FOXO1 in nucleus can enhance its transcription factor activity (DOI: 10.1042/BJ20040167; DOI: 10.15252/embj.2022111867). In our research, neither STAMBPL1 nor its mutant of deubiquitinating enzyme site affected the expression of FOXO1 (Fig.S5E), but STAMBPL1 and FOXO1 co-located in the nucleus (Fig.7M), and they interacted with each other (Fig.7N, Fig.S5I-J). Therefore, we speculate that STAMBPL1 interacts with FOXO1 in the nucleus, obstructs the binding of FOXO1 with the members of 14-3-3 family, inhibits the export of FOXO1, thereby enhancing its transcriptional activity. This interaction between STAMBPL1 and FOXO1 does not necessarily affect the binding of FOXO1 with DNA, including the GRHL3 promoter.

(7) Figure 8 A-C: What is the correlation among the expressions of STAMBPL1, FOXO1, and GRHL3 in TNBC tumors compared to non-TNBC tumors?

Thank you for your comment. In Figure 8A-C, we analyzed the expression levels of *STAMBPL1*, *FOXO1*, and *GRHL3* in both TNBC and non-TNBC samples using the BCIP. The results indicate that the expression levels of these three genes are significantly higher in TNBC compared to non-TNBC samples. To investigate the correlation among the expressions of *STAMBPL1*, *FOXO1*, and *GRHL3* in TNBC versus non-TNBC, we further utilized the Metabric data. Besides the positive correlation trend between *STAMBPL1* and *GRHL3* expression in TNBC clinical samples (Pearson R = 0.27), no significant correlation was observed in the expression levels of *STAMBPL1*, *FOXO1*, and *GRHL3* in TNBC and non-TNBC clinical samples (as shown in Author response image 1 below). Since STAMBPL1 and FOXO1 are involved as protein molecules in the transcriptional regulation of *GRHL3* gene, and the data obtained from the Metabric database are the transcriptional levels of these three genes, this might be the reason why the correlation between their expressions was not observed.

**Author response image 1. sa4fig1:** 

**Reviewer #2 (Recommendations for the authors):**
The authors have thoroughly elucidated the role of STAMBPL1 in TNBC. However, it would be beneficial to discuss the potential clinical implications of these findings, such as how targeting STAMBPL1 or FOXO1 might impact current treatment strategies for TNBC. However, several issues need to be addressed.Major:(1) While the study provides an exhaustive analysis of the molecular mechanisms, a comparison with other subtypes of breast cancer could enhance our understanding of the specificity of the STAMBPL1/FOXO1/GRHL3/HIF1α/VEGFA axis in TNBC.

Thank you for your comment. According to report, STAMBPL1 is significantly associated with the mesenchymal characteristics of breast cancer (DOI: 10.1038/s41416-020-0972-x). We utilized cBioPortal (http://www.cbioportal.org/) to analyze the expression of STAMBPL1 across various clinical subtypes of breast cancer. The results indicated that STAMBPL1 is highly expressed in invasive breast cancer, which has been added to Supplementary Figure 6 as Fig.S6D. Given that TNBC is an aggressive type of invasive breast cancer, we further examined the expression of STAMBPL1 in TNBC compared to non-TNBC using BCIP (http://omicsnet.org/bcancer/database). Our findings revealed that the expression level of STAMBPL1 in TNBC was elevated relative to its levels in non-TNBC (Fig.8A). Additionally, since tumor angiogenesis is a critical factor influencing the metastasis of cancer cells, our study focused specifically on the pro-angiogenic effects of STAMBPL1 in TNBC.

(2) The authors might consider discussing any potential off-target effects of the siRNA and shRNA used in the study to bolster the conclusions drawn from the knockdown experiments.

We appreciate the reviewer's suggestion. It is well-known that siRNA or shRNA have off-target effects. To address this concern, we employed two siRNAs for each gene knockdown in our study. Specifically, we knocked down genes such as *STAMBPL1*, *FOXO1*, *GRHL3*, and *HIF1A* in two TNBC cell lines, HCC1806 and HCC1937, using two siRNAs. Except for siRNA#1 targeting *HIF1A*, which did not show a significant knockdown effect in HCC1806 cells (Fig.2D and Fig.6A), the knockdown effects of other siRNAs on their respective genes were effective, and the resulting phenotypes were consistent. As shown in Fig.2F and Fig.S4H, siRNA#1 targeting *HIF1A* had a significant knockdown effect in HCC1937 cells. The lower knockdown efficiency of this siRNA in HCC1806 cell line might be attributed to cell-specific factors.

(3) It would be advantageous if the authors could provide further details on the patient demographics and tumor characteristics in the TCGA database analysis to better comprehend the clinical relevance of their findings.

Thanks for the reviewer's suggestions. We have now indicated the number of clinical samples in each group in the legend of Fig.8A-C. Since we utilized the BCIP online database to analyze and compare the expression levels of the three genes *STAMBPL1*, *FOXO1*, and *GRHL3* in TNBC and non-TNBC, we are unable to obtain more specific information regarding the tumor characteristics of each sample. However, our analysis clearly shows that the expression levels of these three genes are significantly higher in TNBC compared to non-TNBC.

(4) The authors should consider discussing any limitations regarding the generalizability of their findings, such as potential variations among different TNBC subtypes or the specificity of their observations to certain stages of the disease.

We appreciate the reviewer's comment. Accordingly, we have added a discussion on the limitation of this study in Discussion, highlighted in red font on pages 20 to 21, lines 396 to 412. In addition, we utilized the bc-GenExMiner online database to conduct a comparative analysis of STAMBPL1 expression in different subtypes of non-TNBC and TNBC. The result indicates that STAMBPL1 is highly expressed in mesenchymal-like and basal-like TNBC, which has been added into Supplementary Figure 6 as Fig.S6E. Since these two subtypes of TNBC are highly invasive and metastatic, it suggests that targeting the signaling pathway of STAMBPL1/FOXO1/GRHL3/HIF1α/VEGFA may offer clinical benefits for patients with invasive TNBC.

Minor:The paper is generally well-written, but it's crucial to maintain vigilance for subject-verb agreement, proper use of tense, and consistent terminology.

Thank you for this suggestion. We have thoroughly revised the article for issues such as grammar, including tense, subject-verb agreement, and terminology.